# Privacy Amplification via Random Check-Ins

**Borja Balle**[*]    **Peter Kairouz**[†]    **H. Brendan McMahan**[†]    **Om Thakkar**[†]

**Abhradeep Thakurta**[†]

## Abstract

Differentially Private Stochastic Gradient Descent (DP-SGD) forms a fundamental building block in many applications for learning over sensitive data. Two standard approaches, privacy amplification by subsampling, and privacy amplification by shuffling, permit adding lower noise in DP-SGD than via naïve schemes. A key assumption in both these approaches is that the elements in the data set can be uniformly sampled, or be uniformly permuted — constraints that may become prohibitive when the data is processed in a decentralized or distributed fashion. In this paper, we focus on conducting iterative methods like DP-SGD in the setting of federated learning (FL) wherein the data is distributed among many devices (clients). Our main contribution is the *random check-in* distributed protocol, which crucially relies only on randomized participation decisions made locally and independently by each client. It has privacy/accuracy trade-offs similar to privacy amplification by subsampling/shuffling. However, our method does not require server-initiated communication, or even knowledge of the population size. To our knowledge, this is the first privacy amplification tailored for a distributed learning framework, and it may have broader applicability beyond FL. Along the way, we improve the privacy guarantees of amplification by shuffling and show that, in practical regimes, this improvement allows for similar privacy and utility using data from an order of magnitude fewer users.

## 1   Introduction

Modern mobile devices and web services benefit significantly from large-scale machine learning, often involving training on user (client) data. When such data is sensitive, steps must be taken to ensure privacy, and a formal guarantee of differential privacy (DP) [16, 15] is the gold standard. For this reason, DP has been adopted by companies including Google [20, 10, 18], Apple [2], Microsoft [13], and LinkedIn [31], as well as the US Census Bureau [26].

Other privacy-enhancing techniques can be combined with DP to obtain additional benefits. In particular, cross-device federated learning (FL) [27] allows model training while keeping client data decentralized (each participating device keeps its own local dataset, and only sends model updates or gradients to the coordinating server). However, existing approaches to combining FL and DP make a number of assumptions that are unrealistic in real-world FL deployments such as [11]. To highlight these challenges, we must first review the state-of-the-art in centralized DP training, where differentially private stochastic gradient descent (DP-SGD) [33, 9, 1] is ubiquitous. It achieves optimal error for convex problems [9], and can also be applied to non-convex problems, including deep learning, where the privacy amplification offered by randomly subsampling data to form batches is critical for obtaining meaningful DP guarantees under computational constraints [25, 9, 1, 5, 35].

---

[*]DeepMind. `borja.balle@gmail.com`
[†]Google. {`kairouz, mcmahan, omthkkr, athakurta`}`@google.com`

Attempts to combine FL and the above lines of DP research have been made previously; notably, [28, 3] extended the approach of [1] to FL and user-level DP. However, these works and others in the area sidestep a critical issue: the DP guarantees require very specific sampling or shuffling schemes assuming, for example, that each client participates in each iteration with a fixed probability. While possible in theory, such schemes are incompatible with the practical constraints and design goals of cross-device FL protocols [11]; to quote [23], a comprehensive recent FL survey, *"such a sampling procedure is nearly impossible in practice."*[3] The fundamental challenge is that clients decide when they will be available for training and when they will check in to the server, and by design the server cannot index specific clients. In fact, it may not even know the size of the participating population.

Our work targets these challenges. Our primary goal is to provide strong central DP guarantees for the final model released by FL-like protocols, under the assumption of a trusted[4] orchestrating server. This is accomplished by building upon recent work on amplification by shuffling [19, 12, 18, 22, 6] and combining it with new analysis techniques targeting FL-specific challenges (e.g., client-initiated communications, non-addressable global population, and constrained client availability).

We propose the first privacy amplification analysis specifically tailored for distributed learning frameworks. At the heart of our result is a novel technique, called *random check-in*, that relies only on randomness independently generated by each individual client participating in the training procedure. We show that distributed learning protocols based on random check-ins can attain privacy gains similar to privacy amplification by subsampling/shuffling (see Table 1 for a comparison), while requiring minimal coordination from the server. While we restrict our exposition to distributed DP-SGD within the FL framework for clarity and concreteness (see Figure 1 for a schematic of one of our protocols), we note that the techniques used in our analyses are broadly applicable to any distributed iterative method and might be of interest in other applications[5].

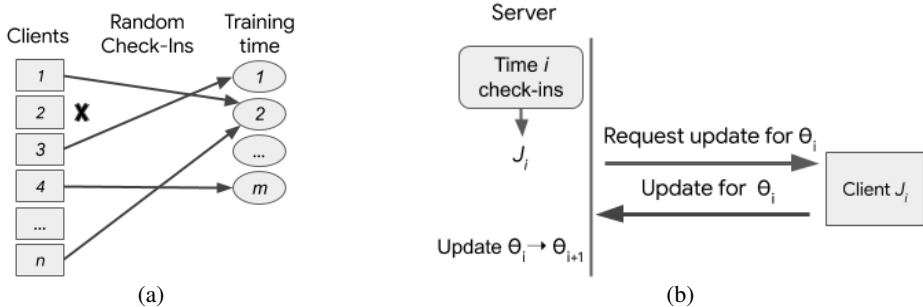

(a)  (b)

Figure 1: A schematic of the Random Check-ins protocol with Fixed Windows (Section 3.1) for Distributed DP-SGD (Algorithm 1). For the central DP guarantee, all solid arrows represent communication over privileged channels not accessible to any external adversary. (a) $n$ clients performing random check-ins with a fixed window of $m$ time steps. 'X' denotes that the client randomly chose to abstain from participating. (b) A time step at the server, where for training time $i \in [m]$, the server selects a client $j$ from those who checked-in for time $i$, requests an update for model $\theta_i$, and then updates the model to $\theta_{i+1}$ (or gradient accumulator if using minibatches).

**Contributions**    The main contributions of this paper can be summarized as follows:

1. We propose *random check-ins*, the first privacy amplification technique for distributed systems with minimal server-side overhead. We also instantiate three distributed learning protocols that use random check-ins, each addressing different natural constraints that arise in applications.

2. We provide formal privacy guarantees for our protocols, and show that random check-ins attain similar rates of privacy amplification as subsampling and shuffling while reducing the need for

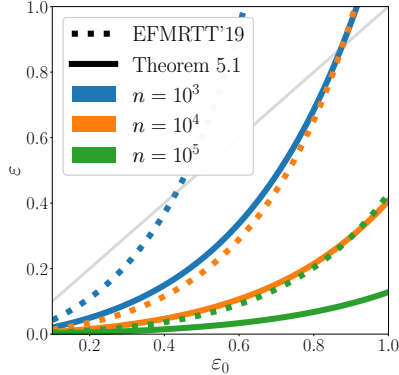

Figure 2: Values of $\varepsilon$ (for $\delta = 10^{-6}$) after amplification by shuffling of $\varepsilon_0$-DP local randomizers obtained from: Theorem 5.1 (solid lines) and [19, Theorem 7] (dotted lines). The grey line represents the threshold of no amplification ($\varepsilon = \varepsilon_0$); after crossing the line amplification bounds become vacuous. Observe that our bounds with $n = 10^3$ and $n = 10^4$ are similar to the bounds from [19] with $n = 10^4$ and $n = 10^5$, respectively.

server-side orchestration. We also provide utility guarantees for one of our protocols in the convex case that match the optimal privacy/accuracy trade-offs for DP-SGD in the central setting [8].

3. As a byproduct of our analysis, we improve privacy amplification by shuffling [19] on two fronts. For $\varepsilon_0$-DP local randomizers, we improve the dependency of the final central DP $\varepsilon$ by $O(e^{0.5\varepsilon_0})$. Figure 2 provides a numerical comparison of the bound from [19] with our bound; for typical parameter values this improvement allows us to provide similar privacy guarantees while reducing the number of required users by one order of magnitude. We also extend the analysis to the case of $(\varepsilon_0, \delta_0)$-DP local randomizers, including Gaussian randomizers that are widely used in practice.

**Related work**  Our work considers the paradigm of federated learning as a stylized example throughout the paper. We refer the reader to [23] for an excellent overview of the state-of-the-art in federated learning, along with a suite of interesting open problems. There is a rich literature on studying differentially private ERM via DP-SGD [33, 9, 1, 36, 34, 30]. However, constraints such as limited availability in distributed settings restrict direct applications of existing techniques. There is also a growing line of works on privacy amplification by shuffling [10, 19, 12, 4, 6, 22, 18] that focus on various ways in which protocols can be designed using trusted shuffling primitives. Lastly, privacy amplification by iteration [21] is another recent advancement that can be applied in an iterative distributed setting, but it is limited to convex objectives.

## 2  Background and Problem Formulation

**Differential Privacy**  We first define neighboring data sets. We refer to a pair of data sets $D, D' \in \mathcal{D}^n$ as neighbors if $D'$ can be obtained from $D$ by modifying one sample $d_i \in D$ for some $i \in [n]$.

**Definition 2.1** (Differential privacy [16, 15])**.** *A randomized algorithm $\mathcal{A} : \mathcal{D}^n \to \mathcal{S}$ is $(\varepsilon, \delta)$-differentially private if, for any pair of neighboring data sets $D, D' \in \mathcal{D}^n$, and for all events $S \subseteq \mathcal{S}$ in the output range of $\mathcal{A}$, we have $\mathbf{Pr}[\mathcal{A}(D) \in S] \leq e^\varepsilon \cdot \mathbf{Pr}[\mathcal{A}(D') \in S] + \delta$.*

For meaningful *central DP* guarantees (i.e., when $n > 1$), $\varepsilon$ is assumed to be a small constant, and $\delta \ll 1/n$. The case $\delta = 0$ is often referred to as *pure DP* (in which case, we just write $\varepsilon$-DP). Specializing Definition 2.1 to the case $n = 1$ gives what we call a *local randomizer*, which provides a *local DP* guarantee. Local randomizers are the typical building blocks of local DP protocols where individuals privatize their data before sending it to an aggregator for analysis [25].

Adaptive DP mechanisms occur naturally when constructing complex DP algorithms, for e.g., DP-SGD. In addition to the dataset $D$, adaptive mechanisms also receive as input the output of other DP mechanisms. Formally, we say that an adaptive mechanism $\mathcal{A} : \mathcal{S}' \times \mathcal{D}^n \to \mathcal{S}$ is $(\varepsilon, \delta)$-DP if the mechanism $\mathcal{A}(s', \bullet)$ is $(\varepsilon, \delta)$-DP for every $s' \in \mathcal{S}'$.

**Problem Setup**  The distributed learning setup we consider here involves $n$ clients, where each client $j \in [n]$ holds a data record[6] $d_j \in \mathcal{D}$, $j \in [n]$, forming a distributed data set $D = (d_1, \ldots, d_n)$. We assume a coordinating server wants to train the parameters $\theta \in \Theta$ of a model by using the dataset

$D$ to perform stochastic gradient descent steps according to some loss function $\ell : \mathcal{D} \times \Theta \to \mathbb{R}_+$. The server's goal is to protect the privacy of all the individuals in $D$ by providing strong DP guarantees against an adversary that can observe the final trained model as well as all the intermediate model parameters. We assume the server is trusted, all devices are honest-but-curious (i.e., they adhere to the prescribed protocol, and there are no malicious users), and all server-client communications are privileged (i.e., they cannot be detected or eavesdropped by an external adversary).

The server starts with model parameters $\theta_1$ and produces a sequence of model parameters $\theta_2, \ldots, \theta_{m+1}$ over a sequence of $m$ time slots. Our random check-ins technique allows clients to independently decide when to offer their contributions for a model update. If and when a client's contribution is accepted by the server, she uses the current parameters $\theta$ and her data $d$ to send a privatized gradient of the form $\mathcal{A}_{ldp}(\nabla_\theta \ell(d, \theta))$ to the server, where $\mathcal{A}_{ldp}$ is a DP local randomizer (e.g., performing gradient clipping and adding Gaussian noise [1]).

Our results consider three different setups inspired by practical applications [11]: (1) The server uses $m \ll n$ time slots, where at most one user's update is used in each slot, for a total of $m/b$ minibatch SGD iterations. It is assumed all $n$ users are available for the duration of the protocol, but the server does not have enough bandwidth to process updates from every user (Section 3.1); (2) The server uses $m \approx n/b$ time slots, and all $n$ users are available for the duration of the protocol (Section 4.1). On average, $b$ users contribute updates to each time slot, and so, we take $m$ minibatch SGD steps; (3) As with (2), but each user is only available during a small window of time relative to the duration of the protocol (Section 4.2).

# 3 Distributed Learning with Random Check-Ins

This section presents the *random check-ins* technique for privacy amplification in the context of distributed learning. We formally define the random check-ins procedure, describe a fully distributed DP-SGD protocol with random check-ins, and analyze its privacy and utility guarantees.

## 3.1 Random Check-Ins with a Fixed Window

Consider the distributed learning setup described in Section 2 where each client is willing to participate in the training procedure as long as their data remains private. To boost the privacy guarantees provided by the local randomizer $\mathcal{A}_{ldp}$, we will let clients volunteer their updates at a *random* time slot of their choosing. This randomization has a similar effect on the uncertainty about the use of an individual's data on a particular update as the one provided by uniform subsampling or shuffling. We formalize this concept using the notion of random check-in, which can be informally expressed as a client in a distributed iterative learning framework randomizing their instant of participation, and determining with some probability whether to participate in the process at all.

**Definition 3.1** (Random check-in). *Let $\mathcal{A}$ be a distributed learning protocol with $m$ check-in time slots. For a set $R_j \subseteq [m]$ and probability $p_j \in [0, 1]$, client $j$ performs an $(R_j, p_j)$-check-in in the protocol if with probability $p_j$ she requests the server to participate in $\mathcal{A}$ at time step $I \xleftarrow{u.a.r.} R_j$, and otherwise abstains from participating. If $p_j = 1$, we alternatively denote it as an $R_j$-check-in.*

Our first distributed learning protocol based on random check-ins is presented in Algorithm 1. Client $j$ independently decides in which of the possible time steps (if any) she is willing to participate by performing an $(R_j, p_j)$-check-in. We set $R_j = [m]$ for all $j \in [n]$, and assume[7] all $n$ clients are available throughout the duration of the protocol. On the server side, at each time step $i \in [m]$, a random client $J_i$ among all the ones that checked-in at time $i$ is queried: this client receives the current model $\theta_i$, locally computes a gradient update $\nabla_\theta \ell(d_{J_i}, \theta_i)$ using their data $d_{J_i}$, and returns to the server a privatized version of the gradient obtained using a local randomizer $\mathcal{A}_{ldp}$. Clients checked-in at time $i$ that are not selected do not participate in the training procedure. If at time $i$ no client is available, the server adds a "dummy" gradient to update the model.

## 3.2 Privacy Analysis

From a privacy standpoint, Algorithm 1 shares an important pattern with DP-SGD: each model update uses noisy gradients obtained from a random subset of the population. However, two key factors

**Server-side protocol:**
*parameters:* local randomizer $\mathcal{A}_{ldp}$, number of steps $m$

Initialize model $\theta_1 \in \Theta$
Initialize gradient accumulator $g_1 \leftarrow 0^p$
**for** $i \in [m]$ **do**
   $S_i \leftarrow \{j : \text{User}(j) \text{ checked-in at time } i\}$
   **if** $S_i$ is empty **then**
      $\tilde{g}_i \leftarrow \mathcal{A}_{ldp}(0^p)$      // Dummy gradient
   **else**
      Sample $J_i \xleftarrow{u.a.r.} S_i$
      Request $\text{User}(J_i)$ for update to model $\theta_i$
      Receive $\tilde{g}_i$ from $\text{User}(J_i)$
   $(\theta_{i+1}, g_{i+1}) \leftarrow \text{ModelUpdate}(\theta_i, g_i + \tilde{g}_i, i)$
   Output $\theta_{i+1}$

**Client-side protocol for User$(j)$:**
*parameters:* check-in window $R_j$, check-in probability $p_j$, loss function $\ell$, local randomizer $\mathcal{A}_{ldp}$
*private inputs:* datapoint $d_j \in \mathcal{D}$

**if** a $p_j$-biased coin returns heads **then**
   Check-in with the server at time $I \xleftarrow{u.a.r.} R_j$
   **if** receive request for update to model $\theta_I$ **then**
      $\tilde{g}_I \leftarrow \mathcal{A}_{ldp}(\nabla_\theta \ell(d_j, \theta_I))$
      Send $\tilde{g}_I$ to server

**ModelUpdate$(\theta, g, i)$:**
*parameters:* batch size $b$, learning rate $\eta$

**if** $i \mod b = 0$ **then**
   **return** $\left(\theta - \frac{\eta}{b}g, 0^p\right)$    // Gradient descent step
**else**
   **return** $(\theta, g)$          // Skip update

Algorithm 1: $\mathcal{A}_{fix}$ – Distributed DP-SGD with random check-ins (fixed window).

make the privacy analysis of our protocol more challenging than the same via subsampling/shuffling. First, unlike in the case of uniform sampling where the randomness in each update is independent, here there is a correlation induced by the fact that clients that check-in into one step cannot check-in into a different step. Second, in shuffling there is also a similar correlation between updates, but there we can ensure each update uses the same number of datapoints, while here the server does not control the number of clients that will check-in into each individual step. Nonetheless, the following result shows that random check-ins provides privacy amplification comparable to these techniques.

**Theorem 3.2** (Amplification via random check-ins into a fixed window). *Suppose $\mathcal{A}_{ldp}$ is an $\varepsilon_0$-DP local randomizer. Let $\mathcal{A}_{fix} : \mathcal{D}^n \to \Theta^m$ be the protocol from Algorithm 1 with check-in probability $p_j = p_0$ and check-in window $R_j = [m]$ for each client $j \in [n]$. For any $\delta \in (0, 1)$, algorithm $\mathcal{A}_{fix}$ is $(\varepsilon, \delta)$-DP with $\varepsilon = p_0(e^{\varepsilon_0} - 1)\sqrt{\frac{2e^{\varepsilon_0}\log(1/\delta)}{m}} + \frac{p_0^2 e^{\varepsilon_0}(e^{\varepsilon_0}-1)^2}{2m}$. In particular, for $\varepsilon_0 < 1$ and $\delta < 1/100$, we get $\varepsilon \leq 7p_0\varepsilon_0\sqrt{\frac{\log(1/\delta)}{m}}$. Furthermore, if $\mathcal{A}_{ldp}$ is $(\varepsilon_0, \delta_0)$-DP with $\delta_0 \leq \frac{(1-e^{-\varepsilon_0})\delta_1}{4e^{\varepsilon_0}\left(2+\frac{\ln(2/\delta_1)}{\ln(1/(1-e^{-5\varepsilon_0}))}\right)}$, then $\mathcal{A}_{fix}$ is $(\varepsilon', \delta')$-DP with $\varepsilon' = \frac{p_0^2 e^{8\varepsilon_0}(e^{8\varepsilon_0}-1)^2}{2m} + p_0(e^{8\varepsilon_0} - 1)\sqrt{\frac{2e^{8\varepsilon_0}\log(1/\delta)}{m}}$ and $\delta' = \delta + m(e^{\varepsilon'} + 1)\delta_0$.*

**Remark 1** We can always increase privacy in the above statement by decreasing $p_0$. However, this will also increase the number of dummy updates, which suggests choosing $p_0 = \Theta(m/n)$. With such a choice, we obtain an amplification factor of $\sqrt{m}/n$. Critically, however, exact knowledge of the population size is *not* required to have a precise DP guarantee above.

**Remark 2** At first look, the amplification factor of $\sqrt{m}/n$ may appear stronger than the typical $1/\sqrt{n}$ factor obtained via uniform subsampling/shuffling. Note that one run of our technique provides $m$ updates (as opposed to $n$ updates via the other methods). When the server has sufficient capacity, we can set $m = n$ to recover a $1/\sqrt{n}$ amplification. The primary advantage of our approach is that we can benefit from amplification in terms of $n$ even if only a much smaller number of updates are actually processed. We can also extend our approach to recover the $1/\sqrt{n}$ amplification even when the server is rate limited ($p_0 = m/n$), by repeating the protocol $\mathcal{A}_{fix}$ adaptively $n/m$ times to get Corollary 3.3 from Theorem 3.2 and applying advanced composition for DP [17].

**Corollary 3.3.** *For algorithm $\mathcal{A}_{fix} : \mathcal{D}^n \to \Theta^m$ in Theorem 3.2, suppose $\mathcal{A}_{ldp}$ is an $\varepsilon_0$-DP local randomizer s.t. $\varepsilon_0 \leq \frac{2\log(n/8\sqrt{m})}{3}$, and $n \geq (e^{\varepsilon_0} - 1)^2 e^{\varepsilon_0}\sqrt{m}\log(1/\beta)$. Setting $p_0 = \frac{m}{n}$, and running $\frac{n}{m}$ repetitions of $\mathcal{A}_{fix}$ results in a total of $n$ updates, and overall central $(\varepsilon, \delta)$-DP with $\varepsilon = \tilde{O}\left(e^{1.5\varepsilon_0}/\sqrt{n}\right)$ and $\delta \in (0, 1)$, where $\tilde{O}(\cdot)$ hides polylog factors in $1/\beta$ and $1/\delta$.*

**Comparison to Existing Privacy Amplification Techniques** Table 1 provides a comparison of the bound in Corollary 3.3 to other existing techniques, for performing one epoch of training (i.e.,

use one update from each client). For this comparison, we assume that $\varepsilon_0 > 1$, since for $\varepsilon_0 \leq 1$ all the shown amplification bounds can be written as $O\left(\varepsilon_0/\sqrt{n}\right)$. "None" denotes a naïve scheme (with no privacy amplification) where each client is used once in any arbitrary order. Also, in general, the guarantees via privacy amplification by subsampling/shuffling apply only under the assumption of complete participation availability[8] of each client. Thus, they define the upper limits of achieving such amplifications. Lastly, even though the bound in Corollary 3.3 appears better than amplification via shuffling, our technique does include dummy updates which do not occur in the other techniques. For linear optimization problems, it is easy to see that our technique will add a factor of $e$ more noise as compared to the other two amplification techniques at the same privacy level.

| Source of Privacy Amplification | $\varepsilon$ for Central DP |
|---|---|
| None [14, 32] | $\varepsilon_0$ |
| Uniform subsampling [25, 9, 1] | $O\left(e^{\varepsilon_0}/\sqrt{n}\right)$ |
| Shuffling [19] | $O\left(e^{3\varepsilon_0}/\sqrt{n}\right)$ |
| Shuffling (Theorem 5.1, This paper) | $O\left(e^{2.5\varepsilon_0}/\sqrt{n}\right)$ |
| Random check-ins with a fixed window (Theorem 3.2, This paper) | $O\left(e^{1.5\varepsilon_0}/\sqrt{n}\right)$ |

Table 1: Comparison with existing amplification techniques for a data set of size $n$, running $n$ iterations of DP-SGD with batch size of 1 and $\varepsilon_0$-DP local randomizers. For ease of exposition, we assume $(e^{\varepsilon_0} - 1) \approx \varepsilon_0$, and hide polylog factors in $n$ and $1/\delta$.

**Proof Sketch for Theorem 3.2** Here, we provide a summary of the argument[9] used to prove Theorem 3.2 in the case $\delta_0 = 0$. First, note that it is enough to argue about the privacy of the sequence of noisy gradients $\tilde{g}_{1:m}$ by post-processing. Also, the role each client plays in the protocol is symmetric, so w.l.o.g. we can consider two datasets $D, D'$ differing in the first position. Next, we imagine that the last $n-1$ clients make the same random check-in choices in $\mathcal{A}_{fix}(D)$ and $\mathcal{A}_{fix}(D')$. Letting $c_i$ denote the number of such clients that check-in into step $i \in [n]$, we model these choices by a pair of sequences $F = (\bar{d}_{1:m}, w_{1:m})$ where $\bar{d}_i \in \mathcal{D} \cup \{\bot\}$ is the data record of an arbitrary client who checked-in into step $i$ (with $\bot$ representing a "dummy" data record if no client checked-in), and $w_i = 1/(c_i + 1)$ represents the probability that client 1's data will be picked to participate in the protocol at step $i$ if she checks-in in step $i$. Conditioned on these choices, the noisy gradients $\tilde{g}_{1:m}$ produced by $\mathcal{A}_{fix}(D)$ can be obtained by: (1) initializing a dataset $\tilde{D} = (\bar{d}_{1:m})$; (2) sampling $I \xleftarrow{u.a.r.} [m]$, and replacing $\bar{d}_I$ with $d_1$ in $\tilde{D}$ w.p. $p_0 w_I$; (3) producing the outputs $\tilde{g}_{1:m}$ by applying a sequence of $\varepsilon_0$-DP adaptive local randomizers to $\tilde{D} = (\tilde{d}_{1:m})$ by setting $\tilde{g}_i = \mathcal{A}^{(i)}(\tilde{d}_i, \tilde{g}_{1:i-1})$. Here each of the $\mathcal{A}^{(i)}$ uses all past gradients to compute the model $\theta_i$ and return $\tilde{g}_i = \mathcal{A}_{ldp}(\nabla_\theta \ell(\tilde{d}_i, \theta_i))$.

The final step involves a variant of the amplification by swapping technique [19, Theorem 8] which we call amplification by random replacement. The key idea is to reformulate the composition of the $\mathcal{A}^{(i)}$ applied to the random dataset $\tilde{D}$, to a composition of mechanisms of the form $\tilde{g}_i = \mathcal{B}^{(i)}(d_1, F, \tilde{g}_{1:i-1})$. Mechanism $\mathcal{B}^{(i)}$ uses the gradient history to compute $q_i = \mathbf{Pr}[I = i|\tilde{g}_{1:i-1}]$ and returns $\mathcal{A}^{(i)}(d_1, \tilde{g}_{1:i-1})$ with probability $p_0 w_i q_i$, and $\mathcal{A}^{(i)}(\bar{d}_i, \tilde{g}_{1:i-1})$ otherwise. Note that before the process begins, we have $\mathbf{Pr}[I = i] = 1/m$ for every $i$; our analysis shows that the posterior probability after observing the first $i - 1$ gradients is not too far from the prior: $q_i \leq \frac{e^{\varepsilon_0}}{me^{\varepsilon_0} - (e^{\varepsilon_0}-1)(i-1)}$. The desired bound is then obtained by using the overlapping mixtures technique [5] to show that $\mathcal{B}^{(i)}$ is $\log(1 + p_0 q_i(e^{\varepsilon_0} - 1))$-DP with respect to changes on $d_1$, and heterogeneous advanced composition [24] to compute the final $\varepsilon$ of composing the $\mathcal{B}^{(i)}$ adaptively.

### 3.3 Utility Analysis

**Proposition 3.4** (Dummy updates in random check-ins with a fixed window)**.** *For algorithm $\mathcal{A}_{fix}$ : $\mathcal{D}^n \to \Theta^m$ described in Theorem 3.2, the expected number of dummy updates performed by the server is at most $\left(m\left(1 - \frac{p_0}{m}\right)^n\right)$. For $c > 0$ if $p_0 = \frac{cm}{n}$, we get at most $\frac{m}{e^c}$ expected dummy updates.*

**Utility for Convex ERMs** We now instantiate our amplification result (Theorem 3.2) in the context of DP empirical risk minimization (ERM). For convex ERMs, we will show that DP-SGD [33, 9, 1] in conjunction with Theorem 3.2 is capable of achieving optimal privacy/accuracy trade-offs [9].

**Theorem 3.5** (Utility guarantee). *Suppose in algorithm $\mathcal{A}_{fix} : \mathcal{D}^n \to \Theta^m$ described in Theorem 3.2 the loss $\ell : \mathcal{D} \times \Theta \to \mathbb{R}_+$ is L-Lipschitz and convex in its second parameter and the model space $\Theta$ has dimension $p$ and diameter $R$, i.e., $\sup_{\theta, \theta' \in \Theta} \|\theta - \theta'\| \leq R$. Furthermore, let $\mathscr{D}$ be a distribution on $\mathcal{D}$, define the population risk $\mathscr{L}(\mathscr{D}; \theta) = \mathbb{E}_{d \sim \mathscr{D}} [\ell(d; \theta)]$, and let $\theta^* = \arg\min_{\theta \in \Theta} \mathscr{L}(\mathscr{D}; \theta)$. If $\mathcal{A}_{ldp}$ is a local randomizer that adds Gaussian noise with variance $\sigma^2$, and the learning rate for a model update at step $i \in [m]$ is set to be $\eta_i = \frac{R(1 - 2e^{-np_0/m})}{\sqrt{(p\sigma^2 + L^2)i}}$, then the output $\theta_m$ of $\mathcal{A}_{fix}(D)$ on a dataset $D$ containing $n$ i.i.d. samples from $\mathscr{D}$ satisfies[10]*

$$\mathbb{E}_{D, \theta_m} [\mathscr{L}(\mathscr{D}; \theta_m)] - \mathscr{L}(\mathscr{D}; \theta^*) = \widetilde{O}\left( \frac{\sqrt{p\sigma^2 + L^2} \cdot R}{(1 - 2e^{-np_0/m})\sqrt{m}} \right).$$

**Remark 3** Note that as $m \to n$, it is easy to see for $p_0 = \Omega\left(\frac{m}{n}\right)$ that Theorem 3.5 achieves the optimal population risk trade-off [9, 8].

## 4 Variations: Thrifty Updates, and Sliding Windows

This section presents two variants of the main protocol from the previous section. The first variant makes a better use of the updates provided by each user at the expense of a small increase in the privacy cost. The second variant allows users to check-in into a sliding window to model the case where different users might be available during different time windows.

### 4.1 Leveraging Updates from Multiple Users

Now, we present a variant of Algorithm 1 which, at the expense of a mild increase in the privacy cost, removes the need for dummy updates, and for discarding all but one of the clients checked-in at every time step. The server-side protocol of this version is given in Algorithm 2 (the client-side protocol is identical as Algorithm 1). Here, if no client checked-in at some step $i \in [m]$, the server skips the update. Furthermore, if at some step multiple clients checked in, the server requests gradients from all the clients, and performs an update using the average of the submitted noisy gradients.

These changes have the advantage of reducing the noise in the model from dummy updates, and increasing the algorithm's data efficiency by using gradients provided by all available clients. The privacy analysis here becomes more challenging as (1) the adversary gains information about the time steps where no clients checked-in, and (2) the server uses the potentially non-private count $|S_i|$ of clients checked-in at time $i$ when performing the model update. Nonetheless, we show that the privacy guarantees of Algorithm 2 are similar to those of Algorithm 1 with an additional $O(e^{3\varepsilon_0/2})$ factor, and the restriction of non-collusion among the participating clients. For simplicity, we only analyze the case where each client has check-in probability $p_j = 1$.

**Server-side protocol:**
*parameters:* total update steps $m$

Initialize model $\theta_1 \in \mathbb{R}^p$
**for** $i \in [m]$ **do**
    $S_i \leftarrow \{j : \text{User}(j) \text{ checks-in for index } i\}$

    **if** $S_i$ is empty **then**
        $\theta_{i+1} \leftarrow \theta_i$
    **else**
        $\tilde{g}_i \leftarrow 0$
        **for** $j \in S_i$ **do**
            Request User$(j)$ for update to model $\theta_i$
            Receive $\tilde{g}_{i,j}$ from User$(j)$
            $\tilde{g}_i \leftarrow \tilde{g}_i + \tilde{g}_{i,j}$
        $\theta_{i+1} \leftarrow \theta_i - \frac{\eta}{|S_i|} \tilde{g}_i$
    Output $\theta_{i+1}$

---

Algorithm 2: $\mathcal{A}_{avg}$ - Distributed DP-SGD with random check-ins (averaged updates).

**Theorem 4.1** (Amplification via random check-ins with averaged updates). *Suppose $\mathcal{A}_{ldp}$ is an $\varepsilon_0$-DP local randomizer. Let $\mathcal{A}_{avg} : \mathcal{D}^n \to \Theta^m$ be the protocol from Algorithm 2 performing $m$ averaged model updates with check-in probability $p_j = 1$ and check-in window $R_j = [m]$ for each user $j \in [n]$. Algorithm $\mathcal{A}_{avg}$ is $(\varepsilon, \delta + \delta_2)$-DP with $\varepsilon = \frac{e^{4\varepsilon_0}(e^{\varepsilon_0}-1)^2\varepsilon_1^2}{2} + e^{2\varepsilon_0}(e^{\varepsilon_0} - 1)\varepsilon_1\sqrt{2\log(1/\delta)}$, where $\varepsilon_1 = \sqrt{\frac{1}{n} + \frac{1}{m}} + \sqrt{\frac{\log(1/\delta_2)}{n}}$. In particular, for $\varepsilon_0 \leq 1$ we get $\varepsilon = O\left(\frac{\varepsilon_0}{\sqrt{m}}\right)$. Furthermore, if $\mathcal{A}_{ldp}$ is $(\varepsilon_0, \delta_0)$-DP with $\delta_0 \leq \frac{(1-e^{-\varepsilon_0})\delta_1}{4e^{\varepsilon_0}\left(2 + \frac{\ln(2/\delta_1)}{\ln(1/(1-e^{-5\varepsilon_0}))}\right)}$, then $\mathcal{A}_{avg}$ is $(\varepsilon', \delta')$-DP with $\varepsilon' = \frac{e^{32\varepsilon_0}(e^{8\varepsilon_0}-1)^2\varepsilon_1^2}{2} + e^{16\varepsilon_0}(e^{8\varepsilon_0}-1)\varepsilon_1\sqrt{2\log\left(\frac{1}{\delta}\right)}$ and $\delta' = \delta + \delta_2 + m(e^{\varepsilon'}+1)\delta_1$.*

We provide a utility guarantee for $\mathcal{A}_{avg}$ in terms of the excess population risk for convex ERMs (similar to Theorem 3.5).

**Theorem 4.2** (Utility guarantee of Algorithm 2). *Suppose in algorithm $\mathcal{A}_{avg} : \mathcal{D}^n \to \Theta^m$ described in Theorem 4.1 the loss $\ell : \mathcal{D} \times \Theta \to \mathbb{R}_+$ is $L$-Lipschitz and convex in its second parameter and the model space $\Theta$ has dimension $p$ and diameter $R$, i.e., $\sup_{\theta,\theta' \in \Theta} \|\theta - \theta'\| \leq R$. Furthermore, let $\mathcal{D}$ be a distribution on $\mathcal{D}$, define the population risk $\mathscr{L}(\mathscr{D}; \theta) = \mathbb{E}_{d \sim \mathscr{D}}[\ell(d; \theta)]$, and let $\theta^* = \arg\min_{\theta \in \Theta} \mathscr{L}(\mathscr{D}; \theta)$. If $\mathcal{A}_{ldp}$ is a local randomizer that adds Gaussian noise with variance $\sigma^2$, and the learning rate for a model update at step $i \in [m]$ is set to be $\eta_i = \frac{R\sqrt{n}}{\sqrt{(mp\sigma^2 + nL^2)i}}$, then the output $\theta_m$ of $\mathcal{A}_{avg}(D)$ on a dataset $D$ containing $n$ i.i.d. samples from $\mathscr{D}$ satisfies*

$$\mathbb{E}_{D,\theta_m}[\mathscr{L}(\mathscr{D}; \theta_m)] - \mathscr{L}(\mathscr{D}; \theta^*) = \widetilde{O}\left(\frac{R\sqrt{mp\sigma^2 + nL^2}}{\sqrt{mn}}\right).$$

*Furthermore, if the loss $\ell$ is $\beta$-smooth in its second parameter and we set the step-size $\eta_i = \frac{R\sqrt{n}}{\beta R\sqrt{n} + m\sqrt{L^2 + p\sigma^2}}$, then we have*

$$\mathbb{E}_{D,\theta_1,\ldots,\theta_m}\left[\mathscr{L}\left(\mathscr{D}; \frac{1}{m}\sum_{i=1}^m \theta_i\right)\right] - \mathscr{L}(\mathscr{D}; \theta^*) = \widetilde{O}\left(R\sqrt{\frac{L^2 + p\sigma^2}{n}} + \frac{\beta R^2}{m}\right).$$

**Comparison of the utility of Algorithm 2 to that of Algorithm 1:** Recall that in $\mathcal{A}_{fix}$ we can achieve a small fixed $\varepsilon$ by taking $p_0 = m/n$ and $\sigma = \tilde{O}(p_0/\varepsilon\sqrt{m})$, in which case the excess risk bound in Theorem 3.5 becomes $\tilde{O}\left(\sqrt{\frac{L^2}{m} + \frac{p}{\varepsilon^2 n^2}}\right)$. On the other hand, in $\mathcal{A}_{avg}$ we can obtain a fixed small $\varepsilon$ by taking $\sigma = \tilde{O}(1/\varepsilon\sqrt{m})$. In this case the excess risks in Theorem 4.2 are bounded by $\tilde{O}\left(\sqrt{\frac{L^2}{m} + \frac{p}{\varepsilon^2 nm}}\right)$ in the convex case, and by $\tilde{O}\left(\sqrt{\frac{L^2}{n} + \frac{p}{\varepsilon^2 nm}} + \frac{1}{m}\right)$ in the convex and smooth case. Thus, we observe that all the bounds recover the optimal population risk trade-offs from [9, 8] as $m \to n$, and for $m \ll n$ and non-smooth loss $\mathcal{A}_{fix}$ provides a better trade-off than $\mathcal{A}_{avg}$, while on smooth losses $\mathcal{A}_{avg}$ and $\mathcal{A}_{fix}$ are incomparable. Note that $\mathcal{A}_{fix}$ (with $b = 1$) will not attain a better bound on smooth losses because each update is based on a single data-point. Setting $b > 1$ will reduce the number of updates to $m/b$ for $\mathcal{A}_{fix}$, whereas to get an excess risk bound for $\mathcal{A}_{fix}$ for smooth losses where more than one data point is sampled at each time step will require extending the privacy analysis to incorporate the change, which is beyond the scope of this paper.

## 4.2 Random Check-Ins with a Sliding Window

The second variant we consider removes the need for all clients to be available throughout the training period. Instead, we assume that the training period comprises of $n$ time steps, and each client $j \in [n]$ is only available during a window of $m$ time steps. Clients perform a random check-in to provide the server with an update during their window of availability. For simplicity, we assume clients wake up in order, one every time step, so client $j \in [n]$ will perform a random check-in within the window $R_j = \{j, \ldots, j + m - 1\}$. The server will perform $n - m + 1$ updates starting at time $m$ to provide a warm-up period where the first $m$ clients perform their random check-ins.

**Theorem 4.3** (Amplification via random check-ins with sliding windows). *Suppose $\mathcal{A}_{ldp}$ is an $\varepsilon_0$-DP local randomizer. Let $\mathcal{A}_{sldw} : \mathcal{D}^n \to \Theta^{n-m+1}$ be the distributed algorithm performing $n$ model updates with check-in probability $p_j = 1$ and check-in window $R_j = \{j, \ldots, j+m-1\}$ for each user $j \in [n]$. For any $m \in [n]$, algorithm $\mathcal{A}_{sldw}$ is $(\varepsilon, \delta)$-DP with $\varepsilon = \frac{e^{\varepsilon_0}(e^{\varepsilon_0}-1)^2}{2m} + (e^{\varepsilon_0} - 1)\sqrt{\frac{2e^{\varepsilon_0}\log(1/\delta)}{m}}$. For $\varepsilon_0 < 1$ and $\delta < 1/100$, we get $\varepsilon \leq 7\varepsilon_0\sqrt{\frac{\log(1/\delta)}{m}}$. Furthermore, if $\mathcal{A}_{ldp}$ is $(\varepsilon_0, \delta_0)$-DP with $\delta_0 \leq \frac{(1-e^{-\varepsilon_0})\delta_1}{4e^{\varepsilon_0}\left(2+\frac{\ln(2/\delta_1)}{\ln(1/(1-e^{-5\varepsilon_0}))}\right)}$, then $\mathcal{A}_{sldw}$ is $(\varepsilon', \delta')$-DP with $\varepsilon' = \frac{e^{8\varepsilon_0}(e^{8\varepsilon_0}-1)^2}{2m} + (e^{8\varepsilon_0} - 1)\sqrt{\frac{2e^{8\varepsilon_0}\log(1/\delta)}{m}}$ and $\delta' = \delta + m(e^{\varepsilon'} + 1)\delta_1$.*

**Remark 4** We can always increase privacy in the statement above by increasing $m$. However, that also increases the number of clients who do not participate in training because their scheduled check-in time is before the process begins, or after it terminates. Moreover, the number of empty slots where the server introduces dummy updates will also increase, which we would want to minimize for good accuracy. Thus, $m$ introduces a trade-off between accuracy and privacy.

**Proposition 4.4** (Dummy updates in random check-ins with sliding windows). *For algorithm $\mathcal{A}_{sldw} : \mathcal{D}^n \to \Theta^{n-m+1}$ described in Theorem 4.3, the expected number of dummy gradient updates performed by the server is at most $(n - m + 1)/e$.*

# 5 Improvements to Amplification via Shuffling

Here, we provide an improvement on privacy amplification by shuffling. This is obtained using two technical lemmas (deferred to the full version [7]) to tighten the analysis of amplification by swapping, a central component in the analysis of amplification by shuffling given in [19].

**Theorem 5.1** (Amplification via Shuffling). *Let $\mathcal{A}^{(i)} : \mathcal{S}^{(1)} \times \cdots \times \mathcal{S}^{(i-1)} \times \mathcal{D} \to \mathcal{S}^{(i)}$, $i \in [n]$, be a sequence of adaptive $\varepsilon_0$-DP local randomizers. Let $\mathcal{A}_{sl} : \mathcal{D}^n \to \mathcal{S}^{(1)} \times \cdots \times \mathcal{S}^{(n)}$ be the algorithm that given a dataset $D = (d_1, \ldots, d_n) \in \mathcal{D}^n$ samples a uniform random permutation $\pi$ over $[n]$, sequentially computes $s_i = \mathcal{A}^{(i)}(s_{1:i-1}, d_{\pi(i)})$ and outputs $s_{1:n}$. For any $\delta \in (0,1)$, algorithm $\mathcal{A}_{sl}$ satisfies $(\varepsilon, \delta)$-DP with $\varepsilon = \frac{e^{3\varepsilon_0}(e^{\varepsilon_0}-1)^2}{2n} + e^{3\varepsilon_0/2}(e^{\varepsilon_0}-1)\sqrt{\frac{2\log(1/\delta)}{n}}$. Furthermore, if $\mathcal{A}^{(i)}$, $i \in [n]$, is $(\varepsilon_0, \delta_0)$-DP with $\delta_0 \leq \frac{(1-e^{-\varepsilon_0})\delta_1}{4e^{\varepsilon_0}\left(2+\frac{\ln(2/\delta_1)}{\ln(1/(1-e^{-5\varepsilon_0}))}\right)}$, then $\mathcal{A}_{sl}$ satisfies $(\varepsilon', \delta')$-DP with*

$$\varepsilon' = \frac{e^{24\varepsilon_0}(e^{8\varepsilon_0}-1)^2}{2n} + e^{12\varepsilon_0}(e^{8\varepsilon_0}-1)\sqrt{\frac{2\log(1/\delta)}{n}} \text{ and } \delta' = \delta + n(e^{\varepsilon'}+1)\delta_1.$$

For comparison, the guarantee in [19, Theorem 7] in the case $\delta_0 = 0$ results in

$$\varepsilon = 2e^{2\varepsilon_0}(e^{\varepsilon_0}-1)\left(e^{\frac{2\exp(2\varepsilon_0)(e^{\varepsilon_0}-1)}{n}} - 1\right) + 2e^{2\varepsilon_0}(e^{\varepsilon_0}-1)\sqrt{\frac{2\log(1/\delta)}{n}}.$$

# 6 Discussion

**Proposed framework's utility for non-convex problems:** Our analytical arguments (Theorems 3.5 and 4.2) formally demonstrate order-optimal utility/privacy trade-offs for convex models. While we agree that this line of research should eventually demonstrate empirical evidence for efficacy, theoretical conclusions do act as guiding principles, and the novel privacy amplification results are interesting and relevant on their own. One can view this line of reasoning to be analogous to privacy-preserving deep learning research, in that the theoretical guarantees for convex models acted as guiding principles for non-convex (deep learning) models [9, 1, 29].

# 7 Broader Impacts

The rapid growth in connectivity and information sharing has been accelerating the adoption of tighter privacy regulations and better privacy-preserving technologies. Therefore, training machine learning models on decentralized data using mechanisms with formal guarantees of privacy is highly desirable. However, despite the rapid acceleration of research on both DP and FL, only a tiny fraction of production ML models are trained using either technology. This work takes an important step in addressing this gap.

Our work highlights the fact that proving DP guarantees for distributed or decentralized systems can be substantially more challenging than for centralized systems, because in the distributed world it becomes much harder to precisely control and characterize the randomness in the system, and this precise characterization and control of randomness is at the heart of DP guarantees. Specifically, production FL systems do not satisfy the assumptions that are typically made under state-of-the-art privacy accounting schemes, such as privacy amplification via subsampling. Without such accounting schemes, service providers cannot give DP statements with small $\varepsilon$'s. This work, though largely theoretical in nature, proposes a method shaped by the practical constraints of distributed systems that allows for rigorous privacy statements under realistic assumptions.

Nevertheless, there is more to do. Our theorems are sharpest in the high-privacy regime (small $\varepsilon$'s), which may be too conservative to provide sufficient utility for some applications. While significantly relaxed from previous work, our assumptions will still not hold in all real-world systems. Thus, we hope this work encourages further collaboration between distributed systems and DP theory researchers in establishing protocols that address the full range of possible systems constraints as well as improving the full breadth of the privacy vs. utility Pareto frontier.

## Acknowledgements

The authors would like to thank Vitaly Feldman for suggesting the idea of privacy accounting in DP-SGD via shuffling, and for help in identifying and fixing a mistake in the way a previous version of this paper handled $(\varepsilon_0, \delta_0)$-DP local randomizers. No third party funding was received by any of the authors to pursue this work.

## Footnotes

[3]In cross-silo FL applications [23], an enumerated set of addressable institutions or data-silos participate in FL, and so explicit server-mediated subsampling or shuffling using existing techniques may be feasible.

[4]Notably, our guarantees are obtained by amplifying the privacy provided by local DP randomizers; we treat this use of local DP as an implementation detail in accomplishing the primary goal of central DP. As a byproduct, our approach offers (weaker) local DP guarantees even in the presence of an untrusted server.

[5]In particular, the Federated Averaging [27] algorithm, which computes an update based on multiple local SGD steps rather than a single gradient, can immediately be plugged into our framework.

[6]Each client is identified as a user. In a general FL setting, each $d_j$ can correspond to a local data set [11].

[7]We make this assumption only for utility; the privacy guarantees are independent of this assumption.

[8]By a complete participation availability for a client, we mean that the client should be available to participate when requested by the server for any time step(s) of training.

[9]Full proofs for every result in the paper are provided in the full version [7].

[10]Here, $\widetilde{O}$ hides a polylog factor in $m$.

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
