[Supplementary Material 1 · Amplification_via_random_check_ins_supp_main.pdf]

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

[Supplementary Material 2 · Amplification_via_random_check_ins_supp.pdf]

## A Omitted Results and Proofs

**Lemma A.1.** *Let $\mathcal{A}_{ldp} : \mathcal{D} \to \mathcal{S}$ be an $\varepsilon_0$-DP local randomizer. For $D = (d_1, \ldots, d_m) \in \mathcal{D}^m, q \in (0,1)$, and $k \in [m]$, define $\mathsf{BiasedSampling}_q(D, k)$ to return $d_k$ with probability $q$, and a sample from an arbitrary distribution over $D \setminus \{d_k\}$ with probability $1 - q$. For any $k \in [m]$ and any set of outcomes $S \subseteq \mathcal{S}$, we have*

$$\frac{\mathbf{Pr}\left[\mathcal{A}_{ldp}(d_k) \in S\right]}{\mathbf{Pr}\left[\mathcal{A}_{ldp}\left(\mathsf{BiasedSampling}_q(D, k)\right) \in S\right]} \leq \frac{e^{\varepsilon_0}}{1 + q(e^{\varepsilon_0} - 1)}.$$

*Proof.* Fix a set of outcomes $S \subseteq \mathcal{S}$. By $\varepsilon_0$-LDP of $\mathcal{A}_{ldp}$, for any $d, d' \in \mathcal{D}$, we get

$$\frac{\mathbf{Pr}[\mathcal{A}_{ldp}(d) \in S]}{\mathbf{Pr}[\mathcal{A}_{ldp}(d') \in S]} \leq e^{\varepsilon_0} \tag{1}$$

Now, for dataset $D = (d_1, \ldots, d_m) \in \mathcal{D}^n$ and $k \in [m]$, we have:

$$\begin{aligned}
\frac{\mathbf{Pr}\left[\mathcal{A}_{ldp}(d_k) \in S\right]}{\mathbf{Pr}\left[\mathcal{A}_{ldp}(\mathsf{BiasedSampling}_q(D, k)) \in S\right]} &= \frac{\mathbf{Pr}\left[\mathcal{A}_{ldp}(d_k) \in S\right]}{\sum\limits_{j=1}^{m} \mathbf{Pr}[\mathcal{A}_{ldp}(d') \in S]\, \mathbf{Pr}[d' = d_j]} \\
&= \frac{1}{\sum\limits_{j=1}^{m} \frac{\mathbf{Pr}[\mathcal{A}_{ldp}(d') \in S]}{\mathbf{Pr}\left[\mathcal{A}_{ldp}(d_k) \in S\right]} \mathbf{Pr}[d' = d_j]} \\
&= \frac{1}{q + \sum\limits_{j \neq k} \frac{\mathbf{Pr}[\mathcal{A}_{ldp}(d') \in S]}{\mathbf{Pr}\left[\mathcal{A}_{ldp}(d_k) \in S\right]} \mathbf{Pr}[d' = d_j]} \\
&\leq \frac{1}{q + e^{-\varepsilon_0} \sum\limits_{j \neq k} \mathbf{Pr}[d' = d_j]} \\
&= \frac{1}{q + (1-q)e^{-\varepsilon_0}} = \frac{e^{\varepsilon_0}}{1 + q(e^{\varepsilon_0} - 1)}
\end{aligned}$$

where the third equality follows as $\mathbf{Pr}[d = d_k] = q$, and the first inequality follows using inequality 1, and the fourth equality follows as $\sum\limits_{j \neq k} \mathbf{Pr}[d = d_j] = 1 - q$. $\qquad\square$

**Lemma A.2.** *Let $\mathcal{A}^{(1)}, \ldots, \mathcal{A}^{(k)}$ be mechanisms of the form $\mathcal{A}^{(i)} : \mathcal{S}^{(1)} \times \cdots \times \mathcal{S}^{(i-1)} \times \mathcal{D} \to \mathcal{S}^{(i)}$. Suppose there exist constants $a > 0$ and $b \in (0,1)$ such that each $\mathcal{A}^{(i)}$ is $\varepsilon_i$-DP with $\varepsilon_i \leq \log\left(1 + \frac{a}{k - b(i-1)}\right)$. Then, for any $\delta \in (0,1)$, the $k$-fold adaptive composition of $\mathcal{A}^{(1)}, \ldots, \mathcal{A}^{(k)}$ is $(\varepsilon, \delta)$-DP with $\varepsilon = \frac{a^2}{2k(1-b)} + \sqrt{\frac{2a^2 \log(1/\delta)}{k(1-b)}}$.*

*Proof.* We start by applying the heterogeneous advanced composition for DP [24] for the sequence of mechanisms $\mathcal{A}_1, \ldots, \mathcal{A}_k$ to get $(\varepsilon, \delta)$-DP for the composition, where

$$\varepsilon = \sum_{i \in [k]} \frac{(e^{\varepsilon_i} - 1)\varepsilon_i}{e^{\varepsilon_i} + 1} + \sqrt{2 \log \frac{1}{\delta} \sum_{i \in [k]} \varepsilon_i^2} \tag{2}$$

Let us start by bounding the second term in equation 2. First, observe that:

$$\sum_{i \in [k]} \varepsilon_i^2 = \sum_{i \in [k]} \left(\log\left(1 + \frac{a}{k - b(i-1)}\right)\right)^2 \leq \sum_{i \in [k]} \frac{a^2}{(k - b(i-1))^2} \tag{3}$$

where the first inequality follows from $\log(1 + x) \leq x$.

Now, we have:

$$\sum_{i \in [k]} \frac{a^2}{(k - b(i-1))^2} = \sum_{i=0}^{k-1} \frac{a^2}{(k - ib)^2} \leq a^2 \int_0^k \frac{1}{(k - xb)^2} dx$$

$$= a^2 \left( \frac{1}{kb - b^2 k} - \frac{1}{kb} \right) = \frac{a^2}{kb} \left( \frac{1}{1-b} - 1 \right)$$

$$= \frac{a^2}{k(1-b)} \tag{4}$$

where the second equality follows as we have $\int \frac{1}{(c-dx)^2} dx = \frac{1}{cd-d^2 x}$.

Next, we bound the first term in equation 2 as follows:

$$\sum_{i \in [k]} \frac{(e^{\varepsilon_i} - 1)\varepsilon_i}{e^{\varepsilon_i} + 1} = \sum_{i \in [k]} \frac{\left( \frac{a}{k-b(i-1)} \right) \left( \log \left( 1 + \frac{a}{k-b(i-1)} \right) \right)}{2 + \frac{a}{k-b(i-1)}} \leq \sum_{i \in [k]} \frac{\left( \frac{a}{k-b(i-1)} \right)^2}{2 + \frac{a}{k-b(i-1)}}$$

$$\leq \sum_{i \in [k]} \frac{a^2}{2 \left( k - b(i-1) \right)^2} \leq \frac{a^2}{2k(1-b)} \tag{5}$$

where the first inequality follows from $\log(1 + x) \leq x$, and the last inequality follows from inequality 4.

Using inequalities 3, 4 and 5 in equation 2, we get that the $k$-fold adaptive composition of $\mathcal{A}_1, \ldots, \mathcal{A}_k$ satisfies $(\varepsilon, \delta)$-DP, for $\varepsilon = \frac{a^2}{2k(1-b)} + \sqrt{\frac{2a^2 \log (1/\delta)}{k(1-b)}}$. □

**Lemma A.3.** *Suppose* $\mathcal{A} : \mathcal{D} \to \mathcal{S}$ *is an* $(\varepsilon_0, \delta_0)$-*DP local randomizer with* $\delta_0 \leq \frac{(1-e^{-\varepsilon_0})\delta_1}{4e^{\varepsilon_0} \left( 2 + \frac{\ln(2/\delta_1)}{\ln(1/(1-e^{-5\varepsilon_0}))} \right)}$. *Then there exists an* $8\varepsilon_0$-*DP local randomizer* $\tilde{\mathcal{A}} : \mathcal{D} \to \mathcal{S}$ *such that for any* $d \in \mathcal{D}$ *we have* $TV(\mathcal{A}(d), \tilde{\mathcal{A}}(d)) \leq \delta_1$.

*Proof.* The proof is a direct application of results by Cheu et al. [12]. First we recall that from [12, Claims D.2 and D.5] (applied with $n = 1$ in their notation) it follows that given $\mathcal{A}$ there exist randomizers $\tilde{\mathcal{A}}_{k,T}$ which are $8\varepsilon_0$-DP and satisfy

$$TV(\mathcal{A}(d), \tilde{\mathcal{A}}_{k,T}(d)) \leq \left( 1 - \frac{ke^{-2\varepsilon_0}}{2} \right)^T + (T+2)\frac{2\delta_0 e^{\varepsilon_0}}{1 - e^{-\varepsilon_0}}$$

for any $k \in (0, 2e^{-2\varepsilon_0})$ and $T \in \mathbb{N}$ as long as $\delta_0 < \frac{1-e^{-\varepsilon_0}}{4e^{\varepsilon_0}}$. The result follows from taking $k = 2e^{-3\varepsilon_0}$, $T = \ln(2/\delta_1)/\ln(1/(1-e^{5\varepsilon_0}))$ and noting these choices imply the desired condition on the total variation distance under our assumption on $\delta_0$. □

*Proof of Corollary 3.3.* Setting $p_0 = \frac{m}{n}$ in $\mathcal{A}_{fix}$, we get from Theorem 3.2 that $\beta \in (0,1)$, algorithm $\mathcal{A}_{fix}$ satisfies $(\varepsilon_1, \beta)$-DP for

$$\varepsilon_1 = \frac{(e^{\varepsilon_0} - 1)\sqrt{2me^{\varepsilon_0} \log (1/\beta)}}{n} + \frac{me^{\varepsilon_0}(e^{\varepsilon_0} - 1)^2}{2n^2}$$

$$\leq \frac{2(e^{\varepsilon_0} - 1)\sqrt{2me^{\varepsilon_0} \log (1/\beta)}}{n} \tag{6}$$

where the inequality follows since $n \geq (e^{\varepsilon_0} - 1)\sqrt{me^{\varepsilon_0}}$.

Now, using inequality 6 and applying advanced composition to $\frac{n}{m}$ repetitions of $\mathcal{A}_{fix}$, we get $\left( \varepsilon, \frac{n\beta}{m} + \delta \right)$-DP, for

$$\varepsilon \leq \varepsilon_1 \sqrt{\frac{2n}{m} \log (1/\delta)} + \frac{n}{m}\varepsilon_1 (e^{\varepsilon_1} - 1) \tag{7}$$

Since $\varepsilon_0 \leq \frac{2 \log \left(n/8\sqrt{m}\right)}{3}$, we have that $\varepsilon_1 \leq \frac{1}{2}$, and thus, $(e^{\varepsilon_1} - 1) \leq \frac{3\varepsilon_1}{2}$. Therefore, we get from inequality 7 that

$$\varepsilon \leq \varepsilon_1 \sqrt{\frac{2n}{m} \log\left(1/\beta\right)} + \frac{3n}{2m} \varepsilon_1^2$$

$$\leq 4(e^{\varepsilon_0} - 1) \sqrt{\frac{e^{\varepsilon_0} \log\left(1/\beta\right) \log\left(1/\delta\right)}{n}} + \frac{12(e^{\varepsilon_0} - 1)^2 e^{\varepsilon_0} \log\left(1/\beta\right)}{n}$$

$$= \widetilde{O}\left(\frac{e^{1.5\varepsilon_0}}{\sqrt{n}}\right)$$

where the equality holds since $n \geq (e^{\varepsilon_0} - 1)^2 e^{\varepsilon_0} \sqrt{m} \log\left(1/\beta\right)$, and $\widetilde{O}(\cdot)$ hides polylog factors in $1/\beta$ and $1/\delta$. $\qquad \square$

*Proof of Proposition 3.4.* In Algorithm 1, for $i \in [m]$, we have

$$S_i = \{j : \text{User}(j) \text{ checks-in for index } i\}$$

For $i \in [m]$, define an indicator random variable $E_i$ that indicates if $S_i$ is empty. Note that the server performs a dummy gradient update for instance $i \in [n]$ if and only if $S_i$ is empty (or, in other words, $E_i = 1$). Next, for $j \in [n]$, let $I_j$ denote the index that user $j$ in Algorithm $\mathcal{A}_{fix}$ performs her $(R_j, p_j)$-check-in into, where $R_j = [m]$ and $p_j = p_0$. Thus, for index $i \in [m]$, we have

$$\mathbf{Pr}\left[E_i = 1\right] = \mathbf{Pr}\left[\bigcap_{j \in [n]} \left((\text{User } j \text{ abstains}) \bigcup (\text{User } j \text{ participates} \wedge I_j \neq i)\right)\right]$$

$$= \prod_{j \in [n]} ((1 - p_0) + \mathbf{Pr}\left[I_j \neq i\right] \cdot p_0) = \left((1 - p_0) + \left(1 - \frac{1}{m}\right) \cdot p_0\right)^n$$

$$= \left(1 - \frac{p_0}{m}\right)^n$$

where the second equality follows since the check-ins for each user are independent of the others, and each user abstains from participating w.p. $(1 - p_0)$.

Thus, for the expected number of dummy gradient updates, we have:

$$\mathbb{E}(E_{1:m}) = \sum_{i \in [m]} \mathbf{Pr}\left[E_i = 1\right] = m\left(1 - \frac{p_0}{m}\right)^n \tag{8}$$

If $p_0 = \frac{cm}{n}$ for $c > 0$, from equation 8 we get

$$\mathbb{E}(E_{1:m}) = m\left(1 - \frac{c}{n}\right)^n \leq \frac{m}{e^c}$$

where the inequality follows as $\left(1 - \frac{a}{b}\right)^b \leq e^{-a}$ for $b > 1, |a| \leq b$. $\qquad \square$

*Proof of Theorem 3.5.* To be able to directly apply [33, Theorem 2], our technique $\mathcal{A}_{fix}$ needs to satisfy two conditions: i) each model update should be an unbiased estimate of the gradient, and ii) a bound on the expected $L_2$-norm of the gradient. Notice that in $\mathcal{A}_{fix}$, every client $j \in [n]$ performs a $([m], p_0)$-check-in. This is analogous to a bins-and-balls setting where $n$ balls are thrown, each with probability $p_0$, into $m$ bins. Thus, for each update step $i \in [m]$, the number of clients checking-in for this step (i.e., $|S_i|$ in the notation of Algorithm 1) can be approximated by an independent Poisson random variable $Y_i$ with mean $np_0/m$, using Poisson approximation [30], as follows:

$$\mathbf{Pr}[|S_i| = 0] \leq 2\,\mathbf{Pr}[Y_i = 0] = 2e^{-np_0/m} := p'$$

Now, we know that there exists a probability $p_b \leq p'$ with which the gradient update $g_i$ is $0^p$. Thus, to make the gradient update unbiased, each participating user can multiply their update by $\frac{1}{1-p_b} \leq \frac{1}{1-p'} = \frac{1}{1-2e^{-np_0/m}}$. Consequently, the Lipschitz-constant of the loss $\ell$, and the variance of the noise added to the update, increases by a factor of at most $\frac{1}{\left(1-2e^{-np_0/m}\right)^2}$. Thus, we get

$\mathbb{E}\left[||\widetilde{g}_i||^2\right] \leq \frac{p\sigma^2 + L^2}{1 - 2e^{-np_0/m}}$. With this, our technique will satisfy both the conditions required to apply the result in [33] for learning rate $\eta_i = \frac{c}{\sqrt{i}}$ as follows:

$$\mathbb{E}_{D,\theta_m}\left[\mathscr{L}(\mathscr{D};\theta_m)\right] - \mathscr{L}(\mathscr{D};\theta^*) \leq \left(\frac{R^2}{c} + \frac{c(p\sigma^2 + L^2)}{1 - 2e^{-np_0/m}}\right)\left(\frac{2 + \log(m)}{\sqrt{m}}\right)$$

Optimizing the learning rate to be $\eta_i = \frac{R\left(1 - 2e^{-np_0/m}\right)}{\sqrt{(p\sigma^2 + L^2)i}}$ gives the statement of the theorem. $\qquad\square$

*Proof of Theorem 4.2.* We prove the first bound on the line of the proof of Theorem 3.5. Since $\mathcal{A}_{avg}$ skips an update for time step $i \in [m]$ if no client checks-in at step $i$, and otherwise makes an update of the average of the noisy gradients received by checked-in clients, each update of the algorithm is unbiased. Now, notice that in $\mathcal{A}_{avg}$, every client $j \in [n]$ checks into $[m]$ u.a.r. Thus, each update step $i \in [m]$ will have $n/m$ checked-in clients in expectation. As a result, for an averaged update $\widetilde{h}_i = \frac{\widetilde{g}_i}{|S_i|}$, we get $\mathbb{E}\left[||\widetilde{h}_i||^2\right] \leq \frac{mp\sigma^2}{n} + L^2$. With this, our technique will satisfy both the conditions required to apply [33, Theorem 2] for learning rate $\eta_i = \frac{c}{\sqrt{i}}$, giving:

$$\mathbb{E}_{D,\theta_m}\left[\mathscr{L}(\mathscr{D};\theta_m)\right] - \mathscr{L}(\mathscr{D};\theta^*) \leq \left(\frac{R^2}{c} + c\left(\frac{mp\sigma^2}{n} + L^2\right)\right)\left(\frac{2 + \log(m)}{\sqrt{m}}\right)$$

Optimizing the learning rate to be $\eta_i = \frac{R\sqrt{n}}{\sqrt{(mp\sigma^2 + nL^2)i}}$ gives the statement of the theorem.

When in addition the loss is $\beta$-smooth we can obtain an improved bound on the expected risk – in this case, for the average parameter vector $\frac{1}{m}\sum_i \theta_i$ – by applying [11, Theorem 6.3]. Let $h_i = \nabla_\theta \mathscr{L}(\mathscr{D};\theta_i)$ be the true gradient on the population loss at each iteration. The cited result says that after $m$ iterations with learning rate $\eta_i = \frac{1}{\beta + \frac{\kappa\sqrt{t}}{\sqrt{2}R}}$ with $\kappa^2 \geq \mathbb{E}[\|h_i - \tilde{h}_i\|^2]$ we get

$$\mathbb{E}_{D,\theta_1,\ldots,\theta_m}\left[\mathscr{L}\left(\mathscr{D};\frac{1}{m}\sum_{i=1}^m \theta_i\right)\right] - \mathscr{L}(\mathscr{D};\theta^*) \leq R\kappa\sqrt{\frac{2}{m}} + \frac{\beta R^2}{m}$$

The result now follows from observing that

$$\mathbb{E}[\|h_i - \tilde{h}_i\|^2] \leq \mathbb{E}_{S\sim\text{Bin}(n,1/m)}\left[\frac{1}{S}(L^2 + p\sigma^2)\middle| S > 0\right] = O\left(\frac{m}{n}(L^2 + p\sigma^2)\right)$$

$\qquad\square$

*Proof of Proposition 4.4.* In Algorithm $\mathcal{A}_{sldw}$, for $i \in [n - m + 1]$, we have

$$S_i = \{j : \text{User}(j) \text{ checks-in for index } i\}$$

For $i \in [n]$, define an indicator random variable $E_i$ that indicates if $S_i$ is empty. Note that the server performs a dummy gradient update for instance $i \in [n]$ if and only if $S_i$ is empty (or, in other words, $E_i = 1$). Next, for $j \in [m]$, let $I_j$ denote the index that user $j$ in Algorithm $\mathcal{A}_{fix}$ performs her $R_j$-check-in into, where $R_j = \{j, \ldots, j + m - 1\}$. Thus, for index $i \in \{m, \ldots, n - m + 1\}$, we have

$$\mathbf{Pr}\left[E_i = 1\right] = \mathbf{Pr}\left[\bigcap_{j\in[i-m+1,i]} I_j \neq i\right] = \prod_{j\in[i-m+1,i]} \mathbf{Pr}\left[I_j \neq i\right] = \left(1 - \frac{1}{m}\right)^m \leq \frac{1}{e} \qquad (9)$$

where the second equality follows since the check-ins for each user are independent of the others, and the inequality follows as $\left(1 - \frac{a}{b}\right)^b \leq e^{-a}$ for $b > 1, |a| \leq b$.

Thus, for the expected number of dummy gradient updates, we have:

$$\mathbb{E}(E_{1:n}) = \sum_{i\in\{m,\ldots,n-m+1\}} \mathbf{Pr}\left[E_i = 1\right] \leq \frac{n - m + 1}{e}$$

where the inequality follows from inequality 9. $\qquad\square$

## A.1 Proof of Theorems 3.2 and 4.3

We will first prove the privacy guarantee of $\mathcal{A}_{fix}$ (Algorithm 1) by reducing it to algorithm $\mathcal{A}_{rep}$ (Algorithm 3) that starts by swapping the first element in the dataset by a given replacement element, randomly chooses a position in the dataset to get replaced by the original first element with a given probability, and then carries out DP-SGD with the local randomizer. W.l.o.g., for simplicity we will define $\mathcal{A}_{rep}$ to update the model for 1-sized minibatches (i.e., update at every time step). It is easy to extend to $b$-sized minibatch updates by accumulating the gradient updates for every $b$ steps and then updating the model.

For the proofs that follow, it will be convenient to define additional notation for denoting distance between distributions. Given 2 distributions $\mu$ and $\mu'$, we denote them as $\mu \cong_{(\varepsilon, \delta)} \mu'$ if they are $(\varepsilon, \delta)$-DP close, i.e., if for all measurable outcomes $S$, we have

$$e^{-\varepsilon}\left(\mu'(S) - \delta\right) \le \mu(S) \le e^{\varepsilon}\mu'(S) + \delta$$

---

**Algorithm 3** $\mathcal{A}_{rep}$: DP-SGD with One Random Replacement

---

**Input:** Dataset $D = d_{1:m}$, local randomizer $\mathcal{A}_{ldp}$.
**Parameters:** Initial model $\theta_1 \in \mathbb{R}^p$, weights $w_{1:m}$ where $w_i \in [0, w_{max}]$ for $i \in [m]$, replacement element $d_r$
1: Sample $I \xleftarrow{u.a.r.} [m]$
2: Let $G \leftarrow (d_r, d_{2:m})$
3: Let $\sigma_I(D) \leftarrow (G_{1:I-1}, z_I, G_{I+1:m})$, where $z_I = \begin{cases} d_1 & \text{with probability } w_I \\ G[I] & \text{otherwise} \end{cases}$
4: **for** $i \in [m]$ **do**
5:      $\tilde{g}_i \leftarrow \mathcal{A}_{ldp}(\theta_i; \sigma_I(D)[i])$
6:      $\theta_{i+1} \leftarrow \theta_i - \eta\tilde{g}_i$
7:      Output $\theta_{i+1}$

---

**Theorem A.4** (Amplification via random replacement). *Suppose $\mathcal{A}_{ldp}$ is an $\varepsilon_0$-DP local randomizer. Let $\mathcal{A}_{rep} : \mathcal{D}^m \to \Theta^m$ be the protocol from Algorithm 3. For any $\delta \in (0, 1)$, algorithm $\mathcal{A}_{rep}$ is $(\varepsilon, \delta)$-DP at index 1 in the central model, where $\varepsilon = \frac{w_{max}^2 e^{\varepsilon_0}(e^{\varepsilon_0}-1)^2}{2m} + w_{max}(e^{\varepsilon_0} - 1)\sqrt{\frac{2e^{\varepsilon_0}\log(1/\delta)}{m}}$. In particular, for $\varepsilon_0 \le 1$ and $\delta \le 1/100$, we get $\varepsilon \le 7w_{max}\varepsilon_0\sqrt{\frac{\log(1/\delta)}{m}}$. Here, initial model $\theta_1 \in \mathbb{R}^p$, weights $w_{max} \in [0, 1]$, $w_i \in [0, w_{max}]$ for every $i \in [m]$, and replacement element $d_r \in [0, 1]$ are parameters to $\mathcal{A}_{rep}$. Furthermore, if $\mathcal{A}_{ldp}$ is an $(\varepsilon_0, \delta_0)$-DP local randomizer with $\delta_0 \le \frac{(1-e^{-\varepsilon_0})\delta_1}{4e^{\varepsilon_0}\left(2 + \frac{\ln(2/\delta_1)}{\ln(1/(1-e^{-5\varepsilon_0}))}\right)}$, then algorithm $\mathcal{A}_{rep}$ is $(\varepsilon', \delta')$-DP at index 1 in the central model, where $\varepsilon' = \frac{w_{max}^2 e^{8\varepsilon_0}(e^{8\varepsilon_0}-1)^2}{2m} + w_{max}(e^{8\varepsilon_0} - 1)\sqrt{\frac{2e^{8\varepsilon_0}\log(1/\delta)}{m}}$ and $\delta' = \delta + m(e^{\varepsilon'} + 1)\delta_1$.*

*Proof.* We start by proving the privacy guarantee of $\mathcal{A}_{rep}$ for the case where the local randomizer $\mathcal{A}_{ldp}$ is $\varepsilon_0$-DP, i.e., for the case where $\delta_0 = 0$. Let us denote the output sequence of $\mathcal{A}_{rep}$ by $Z_2, Z_3, \ldots, Z_{m+1}$. Note that $Z_{2:m+1}$ can be seen as the output of a sequence of $m$ algorithms with conditionally independent randomness: $\mathcal{B}^{(i)}$ for $i \in [m]$ as follows. On input $\theta_{2:i}$ and $D$, $\mathcal{B}^{(i)}$ outputs a random sample from the distribution of $Z_{i+1}|Z_{2:i} = \theta_{2:i}$. The outputs of $\mathcal{B}^{(1)}, \ldots, \mathcal{B}^{(i-1)}$ are given as input to $\mathcal{B}^{(i)}$. Therefore, in order to upper bound the privacy parameters of $\mathcal{A}_{rep}$, we analyze the privacy parameters of $\mathcal{B}^{(1)}, \ldots, \mathcal{B}^{(m)}$ and apply the heterogeneous advanced composition for DP [24].

Next, observe that conditioned on the value of $I$, $Z_{i+1}$ is the output of $\mathcal{A}_{ldp}^{(i)}(\theta_i; d)$ with its internal randomness independent of $Z_{2:i}$. In particular, for $i \ge 2$, one can implement $\mathcal{B}^{(i)}$ as follows. First, sample an index $T$ from the distribution of $I|Z_{2:i} = \theta_{2:i}$. Assign $\tilde{g}_i = \mathcal{A}_{ldp}(\theta_i; d_1)$ w.p. $w_i$ if $T = i$, otherwise let $\tilde{g}_i = \mathcal{A}_{ldp}(\theta_i; d_i)$. For $\mathcal{B}^{(1)}$, we first sample $T$ u.a.r. from $[m]$, and let $\tilde{g}_1 = \mathcal{A}_{ldp}(\theta_1; d_1)$ w.p. $w_1$ if $T = 1$, otherwise let $\tilde{g}_1 = \mathcal{A}_{ldp}(\theta_1; d_r)$. For each $i \in [m]$, algorithm $\mathcal{B}^{(i)}$ outputs $\theta_{i+1} = \theta_i - \eta\tilde{g}_i$.

We now prove that for each $i \in [m]$, $\mathcal{B}^{(i)}$ is $\left(\log\left(1 + \frac{w_{max}e^{\varepsilon_0}(e^{\varepsilon_0}-1)}{i-1+e^{\varepsilon_0}(m-i+1)}\right), 0\right)$-DP at index 1. Let $D = d_{1:m}$ and $D' = (d'_1, d_{2:m})$ be 2 datasets differing in the first element. Let $\theta_{2:i}$ denote the input to $\mathcal{B}^{(i)}$. Let $\mu$ be the probability distribution of $\mathcal{B}^{(i)}(\theta_{2:i}; D)$. Let $\mu_1$ be the distribution of $\mathcal{B}^{(i)}(\theta_{2:i}; D)$ conditioned on $\tilde{g}_i = \mathcal{A}_{ldp}(\theta_{2:i}; d_1)$, and $\mu_0$ be the distribution of $\mathcal{B}^{(i)}(\theta_{2:i}; D)$ conditioned on $\tilde{g}_i = \mathcal{A}_{ldp}(\theta_{2:i}; d_r)$ for $i = 1$, and $\tilde{g}_i = \mathcal{A}_{ldp}(\theta_{2:i}; d_i)$ for $i \geq 2$. Also, denote by $\mu', \mu'_0$, and $\mu'_1$ the corresponding quantities when $\mathcal{B}^{(i)}$ is run on $D'$. Let $q_i$ be the probability that $T = i$ (sampled from $I|Z_{2:i} = \theta_{2:i}$). By definition, $\mu = (1 - q_i w_i)\mu_0 + q_i w_i \mu_1$, as $\mathcal{B}^{(i)}(s_{1:i-1}; D)$ generates output using $\mathcal{A}_{ldp}(\theta_{2:i}; d_1)$ w.p. $w_i$ if $T = i$. Similarly, $\mu' = (1 - q'_i w_i)\mu'_0 + q'_i w_i \mu'_1$ when the input dataset is $D'$.

For $i \in [m]$, we observe that $\mu_0 = \mu'_0$, since in both cases the output is generated by $\mathcal{A}_{ldp}(\theta_1, d_r)$ for $i = 1$, and $\mathcal{A}_{ldp}(\theta_{2:i}; d_i)$ for $i \geq 2$. W.l.o.g. assume that $q_i \geq q'_i$. Thus, we can shift $(q_i - q'_i)w_i$ mass from the first component of the mixture in $\mu'$ to the second component to obtain

$$\mu' = (1 - q_i w_i)\mu_0 + q_i w_i \left(\frac{q'_i}{q_i}\mu'_1 + \left(1 - \frac{q'_i}{q_i}\right)\mu_0\right) = (1 - q_i w_i)\mu_0 + q_i w_i \mu''_1$$

This shows that $\mu$ and $\mu'$ are overlapping mixtures [5]. Now, $\varepsilon_0$-LDP of $\mathcal{A}_{ldp}$ implies $\mu_0 \cong_{(\varepsilon_0, 0)} \mu_1$ and $\mu'_0 \cong_{(\varepsilon_0, 0)} \mu'_1$. Moreover, $\varepsilon_0$-LDP of $\mathcal{A}_{ldp}$ also implies $\mu_1 \cong_{(\varepsilon_0, 0)} \mu'_1$, so by the joint convexity of the relation $\cong_{(\varepsilon_0, 0)}$ we also have $\mu_1 \cong_{(\varepsilon_0, 0)} \mu''_1$. Thus, we can apply Advanced Joint Convexity of overlapping mixtures (Theorem 2 in [5]) to get that

$$\mu \cong_{(\log(1+q_i w_i(e^{\varepsilon_0}-1)), 0)} \mu' \tag{10}$$

We now claim that $q_i \leq \frac{e^{\varepsilon_0}}{i-1+e^{\varepsilon_0}(m-i+1)}$. Observe that for each $D^* \in \{D, D'\}$, conditioning on $T = i$ reduces $\mathcal{A}_{rep}$ to running $\mathcal{A}_{ldp}$ on $\sigma_i(D^*)$. Note that for $j < i$, we have that $\sigma_i(D^*)[1:i-1]$ differs from $\sigma_j(D^*)[1:i-1]$ in at most 1 position, and for $j > i$, we have $\sigma_i(D^*)[1:i-1] = \sigma_j(D^*)[1:i-1]$. Since $\mathbf{Pr}[j \geq i] = \frac{m-i+1}{m}$, by setting $q = \frac{m-i+1}{m}$ in Lemma A.1, we get that

$$\frac{\mathbf{Pr}[Z_{2:i} = \theta_{2:i}|T = i]}{\mathbf{Pr}[Z_{2:i} = \theta_{2:i}]} \leq \frac{e^{\varepsilon_0}}{1 + \frac{(m-i+1)}{m}(e^{\varepsilon_0}-1)} = \frac{me^{\varepsilon_0}}{i-1+e^{\varepsilon_0}(m-i+1)} \tag{11}$$

This immediately implies our claim, since we have

$$q_i = \mathbf{Pr}[T = i|Z_{2:i} = \theta_{2:i}] = \frac{\mathbf{Pr}[Z_{2:i} = \theta_{2:i}|T = i] \cdot \mathbf{Pr}[t = i]}{\mathbf{Pr}[Z_{2:i} = \theta_{2:i}]}$$

$$\leq \frac{e^{\varepsilon_0}}{i-1+e^{\varepsilon_0}(m-i+1)}$$

where the inequality follows from inequality 11, and as $\mathbf{Pr}[T = i] = \frac{1}{m}$.

Substituting the value of $q_i$ in equation 10, and using the fact that $w_i \leq w_{max}$, we get that for each $i \in [m]$, algorithm $\mathcal{B}^{(i)}$ is $(\varepsilon_i, 0)$-DP at index 1, where $\varepsilon_i = \log\left(1 + \frac{w_{max}e^{\varepsilon_0}(e^{\varepsilon_0}-1)}{i-1+e^{\varepsilon_0}(m-i+1)}\right)$.

This can alternatively be written as $\varepsilon_i = \log\left(1 + \frac{w_{max}(e^{\varepsilon_0}-1)}{m-(i-1)\frac{e^{\varepsilon_0}-1}{e^{\varepsilon_0}}}\right)$, and using Lemma A.2 for the sequence of mechanisms $\mathcal{B}^{(1)}, \ldots, \mathcal{B}^{(m)}$ by setting $a = w_{max}(e^{\varepsilon_0}-1)$, $b = \frac{e^{\varepsilon_0}-1}{e^{\varepsilon_0}}$, and $k = m$, we get that algorithm $\mathcal{A}_{rep}$ satisfies $(\varepsilon, \delta)$-DP at index 1, for $\varepsilon = \frac{w_{max}^2 e^{\varepsilon_0}(e^{\varepsilon_0}-1)^2}{2m} + w_{max}(e^{\varepsilon_0}-1)\sqrt{\frac{2e^{\varepsilon_0}\log(1/\delta)}{m}}$.

Now, for the above bound, if $\varepsilon_0 \leq 1$ and $\delta \leq 1/4$, we get that

$$\varepsilon = \frac{w_{max}^2 e^{\varepsilon_0}(e^{\varepsilon_0}-1)^2}{2m} + w_{max}(e^{\varepsilon_0}-1)\sqrt{\frac{2e^{\varepsilon_0}\log(1/\delta)}{m}}$$

$$= \frac{w_{max}e^{0.5\varepsilon_0}(e^{\varepsilon_0}-1)}{\sqrt{m}}\left(\frac{w_{max}e^{0.5\varepsilon_0}(e^{\varepsilon_0}-1)}{2\sqrt{m}} + \sqrt{2\log(1/\delta)}\right)$$

$$\leq \frac{3w_{max}\varepsilon_0}{\sqrt{m}}\left(\frac{3w_{max}\varepsilon_0}{2\sqrt{m}} + \sqrt{2\log(1/\delta)}\right)$$

$$\leq \frac{3w_{max}\varepsilon_0}{\sqrt{m}}\left(\left(\sqrt{2} + \sqrt{1/2}\right)\sqrt{\log(1/\delta)}\right) \leq 7w_{max}\varepsilon_0\sqrt{\frac{\log(1/\delta)}{m}}$$

where the first inequality follows since $e^{0.5\varepsilon_0}(e^{\varepsilon_0} - 1) \leq 3\varepsilon_0$ for $\varepsilon_0 \leq 1$, and the second inequality follows since $\frac{3w_{max}\varepsilon_0}{2\sqrt{m}} \leq \sqrt{\frac{\log\frac{1}{\delta}}{2}}$ for $\delta \leq 1/100$.

Now, we prove the privacy guarantee of $\mathcal{A}_{rep}$ for the more general case where for each $i \in [m]$, the local randomizer $\mathcal{A}_{ldp}$ is $(\varepsilon_0, \delta_0)$-DP. To upper bound the privacy parameters of $\mathcal{A}_{rep}$, we modify the local randomizer to satisfy pure DP, apply the previous analysis, and then account for the difference between the protocols with original and modified randomizers using the total variation distance.

Since $\mathcal{A}_{ldp}$ is $(\varepsilon_0, \delta_0)$-DP with $\delta_0 \leq \frac{(1-e^{-\varepsilon_0})\delta_1}{4e^{\varepsilon_0}\left(2+\frac{\ln(2/\delta_1)}{\ln(1/(1-e^{-5\varepsilon_0}))}\right)}$, we get from Lemma A.3 that there exists a randomizer $\tilde{\mathcal{A}}_{ldp}$ that is $8\varepsilon_0$-DP, and for any data record $d$ and parameter vector $\theta$ satisfies $TV\left(\mathcal{A}_{ldp}(d;\theta), \tilde{\mathcal{A}}_{ldp}(d;\theta)\right) \leq \delta_1$. After replacing every instance of $\mathcal{A}_{ldp}$ in $\mathcal{A}_{rep}$ with $\tilde{\mathcal{A}}_{ldp}$ to obtain $\tilde{\mathcal{A}}_{rep}$, a union bound gives:

$$TV\left(\mathcal{A}_{rep}(D); \tilde{\mathcal{A}}_{rep}(D)\right) \leq m\delta_1 \tag{12}$$

Now, proceeding in a similar manner as in the case of $\varepsilon_0$-DP local randomizers above to see that $\tilde{\mathcal{A}}_{rep}$ using the $8\varepsilon_0$-DP local randomizers $\tilde{\mathcal{A}}_{ldp}$ satisfies $(\varepsilon', \delta)$-DP at index 1 with $\varepsilon' = \frac{w_{max}^2 e^{8\varepsilon_0}(e^{8\varepsilon_0}-1)^2}{2m} + w_{max}(e^{8\varepsilon_0} - 1)\sqrt{\frac{2e^{8\varepsilon_0}\log(1/\delta)}{m}}$. Thus, using Proposition 3 from [39] and inequality 12, we get that $\mathcal{A}_{rep}$ satisfies $(\varepsilon', \delta')$-DP at index 1 with $\delta' = \delta + m(e^{\varepsilon'} + 1)\delta_1$. □

Now, we are ready to prove Theorems 3.2 and 4.3.

*Proof of Theorem 3.2.* Let $D$ and $D'$ be 2 datasets of $n$ users that differ in a user at some index $i^* \in [n]$. Algorithm $\mathcal{A}_{fix}$ can be alternatively seen as follows. The server starts by initializing $F = [0^p]^m$, weights $W = [1]^m$, and for $i \in [m]$, set $S_i = \phi$. For each user $j \in [n]$ s.t. $j \neq i^*$, user $j$ performs a random check-in along with some additional operations. She first samples $I_j$ u.a.r. from $[m]$, and w.p. $p_0$ does the following: she requests the server for model at index $I_j$ (and gets inserted into set $S_{I_j}$ at the server). She also updates $F[I_j] = d_j$ with probability $W[I_j]$, and sets $W[I_j] = \frac{W[I_j]}{W[I_j]+1}$. Next, the server runs $\mathcal{A}_{rep}$ on input dataset $\pi^*(D) = (d_{i^*}, F[2:m])$, with the replacement element $F[1]$, initial model $\theta_1$, and weight parameters set to $W'[1:m]$, where $W'[i] = W[i] \cdot p_0$.

First, notice that in the alternative strategy above, for each of the weights $W[i], i \in [m]$, it always holds that $W[i] = \frac{1}{|S_i|}$. Thus, each weight $W[i], i \in [m]$ is updated to simulate reservoir sampling [37] of size 1 in slot $F[i]$. In other words, updating $F[i] = d$ with probability $W[i]$ for an element $d$ is equivalent to $F[j] \xleftarrow{u.a.r.} S_i$, where $S_i$ is the set containing $d$ and all the elements previously considered for updating $S_i$. As a result, since the first element in $\mathcal{A}_{rep}$ performs a random replacement with weights set to $W'[1:m]$ for its input dataset, it is easy to see that performing a concurrent random check-in for user $i^*$ (as in Algorithm 1) is equivalent to performing a random replacement for her after the check-ins of all the other users.

From our construction, we know that datasets $\pi^*(D)$ and $\pi^*(D')$, which are each of length $m$, differ only in the element with index 1. Moreover, in the alternative strategy above, note that the weights $W'[1:m]$ and the replacement element $F[1]$ input to $\mathcal{A}_{rep}$ are independent of the data of user $i^*$ in the original dataset. Therefore, in the case $\delta_0 = 0$, using Theorem A.4 and setting $w_{max} = p_0$, we get $\mathcal{A}_{rep}(\pi^*(D)) \approxeq_{\varepsilon,\delta} \mathcal{A}_{rep}(\pi^*(D'))$ at index 1, for $\varepsilon = \frac{p^2 e^{\varepsilon_0}(e^{\varepsilon_0}-1)^2}{2m} + \frac{p(e^{\varepsilon_0}-1)\sqrt{2e^{\varepsilon_0}\log(1/\delta)}}{m}$, which implies $\mathcal{A}_{dist}(D) \approxeq_{\varepsilon,\delta} \mathcal{A}_{dist}(D')$. Consequently, it implies $\varepsilon \leq 7p_0\varepsilon_0\sqrt{\frac{\log(1/\delta)}{m}}$ for $\varepsilon_0 < 1$ and $\delta < 1/100$.

The case $\delta_0 > 0$ follows from the same reduction using the corresponding setting of Theorem A.4. □

*Proof of Theorem 4.3.* We proceed similar to the proof of Theorem 3.2. Let $D$ and $D'$ be 2 datasets of $n$ users that differ in a user at some index $i^* \in [n]$. Algorithm $\mathcal{A}_{sldw}$ can be alternatively seen

as follows. The server starts by initializing $F = [0^p]^{n-m+1}$, weights $W = [1]^{n-m+1}$, and for $j \in \{m, \ldots, n\}$, set $S_j = \phi$. For each user $j \in [n]$ s.t. $j \neq i^*$, user $j$ performs a random check-in along with some additional operations. She first samples $I_j$ u.a.r. from $\{j, \ldots, j+m-1\}$, requests the server for model at index $I_j$ (and gets inserted into set $S_{I_j}$ at the server). She also updates $F[I_j] = d_j$ with probability $W[I_j]$, and sets $W[I_j] = \frac{W[I_j]}{W[I_j]+1}$.

Now, the server runs its loop until it releases $i^* - 1$ outputs. Next, the server runs $\mathcal{A}_{rep}$ on input dataset $\pi^*(D) = (d_{i^*}, F[i^*+1 : i^*+m])$, with weight parameters set to $W[i^* : i^*+m]$, initializing model $\theta_{i^*}$, and the replacement element $F[i^*]$. Lastly, the server releases the last $(n - (i^* + m) + 1)$ outputs of $\mathcal{A}_{sldw}$ using $F[i^* + m + 1 : n]$ and the local randomizer $\mathcal{A}_{ldp}$.

First, notice that in the alternative strategy above, for each of the weights $W[i], i \in [n]$, it always holds that $W[i] = \frac{1}{|S_i|}$. Thus, each weight $W[i], i \in [n]$ is updated to simulate reservoir sampling [37] of size 1 in slot $F[i]$. In other words, updating $F[i] = d$ with probability $W[i]$ for an element $d$ is equivalent to $F[i] \xleftarrow{u.a.r.} S_i$, where $S_i$ is the set containing $z$ and all the elements previously considered for updating $S_i$. As a result, since the first element in $\mathcal{A}_{rep}$ performs a random replacement for its input dataset (which doesn't include $F_{1:i^*-1} \bigcup F_{i^*+m+1:n}$ in the alternative strategy above), it is easy to see that sequentially performing a random check-in for user $i^*$ (as in Algorithm 1) is equivalent to performing a random replacement for her after the check-ins of all the other users and releasing the first $i^* - 1$ outputs of $\mathcal{A}_{sldw}$.

From our construction, we know that datasets $\pi^*(D)$ and $\pi^*(D')$, which are each of length $m$, differ only in the element with index 1. Moreover, in the alternative strategy above, note that the weights $W[i^* : i^* + m]$, initializing model $\theta_{i^*}$ and the replacement element $F[i^*]$ input to $\mathcal{A}_{rep}$ are independent of the data of user $i^*$ in the original dataset. Therefore, using Theorem A.4 and setting $w_{max} = 1$, we get $\mathcal{A}_{rep}(\pi^*(D)) \cong_{\varepsilon, \delta+m\delta_0} \mathcal{A}_{rep}(\pi^*(D'))$ at index 1, for $\varepsilon = \frac{e^{\varepsilon_0}(e^{\varepsilon_0}-1)^2}{2m} + (e^{\varepsilon_0} - 1)\sqrt{\frac{2e^{\varepsilon_0}\log(1/\delta)}{m}}$, which implies $\mathcal{A}_{rc}(D) \cong_{\varepsilon, \delta+m\delta_0} \mathcal{A}_{rc}(D')$. Consequently, it implies $\varepsilon \leq 7\varepsilon_0\sqrt{\frac{\log(1/\delta)}{m}}$ for $\varepsilon_0 < 1$ and $\delta < 1/100$.

The case $\delta_0 > 0$ follows from the same reduction using the corresponding setting of Theorem A.4.
□

## A.2 Proof of Theorem 4.1

Let $L = (L_1, \ldots, L_m)$ represent the number of users contributing to each of the update steps, i.e., $L_i = |S_i|$ for $i \in [m]$. We start by considering the output distribution of $\mathcal{A}_{avg}(D)$ conditioned on $L = \ell$ for some $\ell \in [n]^m$ s.t. $\sum_i \ell_i = n$. This distribution is the same as the one produced by Algorithm 4 with bin sizes $\ell$ on a random permutation $\pi(D)$ of the original dataset $D$. To analyze the privacy of $\mathcal{A}_{bin}(\pi(D), \ell)$ we use the reduction from shuffling to swapping [19] . This reduction says it suffices to analyze the privacy of $D \mapsto \mathcal{A}_{bin}(\sigma(D), \ell)$ on a pair of datasets $D$ and $D'$ differing in the first record, where $\sigma(D)$ randomly swaps $d_1$ with $d_I$ for $I$ uniformly sampled from $[n]$.

---

**Algorithm 4** $\mathcal{A}_{bin}$: DP-SGD with Bins

---

    **Input:** Dataset $D = d_{1:n}$, bin sizes $\ell \in [n]^m$ with $\sum_i \ell_i = n$, local randomizer $\mathcal{A}_{ldp}$
1: Initialize model $\theta_1 \in \mathbb{R}^p$
2: $j \leftarrow 1$
3: **for** $i \in [m]$ **do**
4:     **if** $\ell_i = 0$ **then**
5:         $\theta_{i+1} \leftarrow \theta_i$
6:     **else**
7:         $\tilde{g}_i \leftarrow 0$
8:         **for** $k \in \{j, \ldots, j+\ell_i-1\}$ **do**
9:             $\tilde{g}_i \leftarrow \tilde{g}_i + \mathcal{A}_{ldp}(d_k, \theta_i)$
10:         $j \leftarrow j + \ell_i$
11:         $\theta_{i+1} \leftarrow \text{ModelUpdate}(\theta_i; \tilde{g}_i/\ell_i)$
12: **return** sequence $\theta_{2:m+1}$

---

**Theorem A.5.** *Suppose $\mathcal{A}_{ldp} : \mathcal{D} \times \Theta \to \Theta$ is an $\varepsilon_0$-DP local randomizer. Let $\ell \in [m]^n$ with $\sum_i \ell_i = n$. Also, for any dataset $D = \{d_1, \ldots, d_n\}$, define $\sigma(D)$ be the operation that randomly swaps $d_1$ with $d_I$ for $I$ uniformly sampled from $[n]$. For any $\delta \in (0,1)$, the mechanism $M(D) = \mathcal{A}_{bin}(\sigma(D), \ell)$ is $(\varepsilon, \delta)$-DP at index 1 with $\varepsilon = \frac{\|\ell\|_2^2 e^{4\varepsilon_0}(e^{\varepsilon_0}-1)^2}{2n^2} + \frac{\|\ell\|_2 e^{2\varepsilon_0}(e^{\varepsilon_0}-1)}{n} \sqrt{2\log(1/\delta)}$. Furthermore, if $\mathcal{A}_{ldp}$ is $(\varepsilon_0, \delta_0)$-DP with $\delta_0 \leq \frac{(1-e^{-\varepsilon_0})\delta_1}{4e^{\varepsilon_0}\left(2 + \frac{\ln(2/\delta_1)}{\ln(1/(1-e^{-5\varepsilon_0}))}\right)}$, then $M$ is $(\varepsilon', \delta')$-DP with*

$$\varepsilon' = \frac{\|\ell\|_2^2 e^{32\varepsilon_0}(e^{8\varepsilon_0}-1)^2}{2n^2} + \frac{\|\ell\|_2 e^{16\varepsilon_0}(e^{8\varepsilon_0}-1)}{n} \sqrt{2\log(1/\delta)} \text{ and } \delta' = \delta + m(e^{\varepsilon'}+1)\delta_1.$$

*Proof.* Let $\sigma(D) = (\tilde{d}_1, \ldots, \tilde{d}_n)$ denote the dataset after the swap operation. Using the bin sizes $\ell$, we split this dataset into $m_0$ disjoint datasets $\tilde{D}_1, \ldots, \tilde{D}_{m_0}$ of sizes $|\tilde{D}_i| = \ell_i$ with $\tilde{D}_1 = (\tilde{d}_1, \ldots, \tilde{d}_{\ell_1})$, and so on. Note that each of the outputs is obtained as $\theta_{i+1} \leftarrow \mathcal{A}^{(i)}(\theta_i; \tilde{D}_i)$ with

$$\mathcal{A}^{(i)}(\theta_i; \tilde{D}_i) = \text{ModelUpdate}\left(\theta_i; \frac{1}{\ell_i} \sum_{\tilde{d} \in \tilde{D}_i} \mathcal{A}_{ldp}(\tilde{d}, \theta_i)\right)$$

By post-processing, each of the $\mathcal{A}^{(i)}$ is $(\varepsilon_0, \delta_0)$-DP.

The next step is to modify these mechanisms to reduce the analysis to a question about adaptive composition. Thus, we introduce mechanisms $\mathcal{B}^{(i)}$ for $i \in [m_0]$ that take as input the whole dataset $D$ and the outputs $\theta_{1:i} = (\theta_1, \ldots, \theta_i)$ of the previous mechanisms. Mechanism $\mathcal{B}^{(i)}$ starts by splitting the dataset $D$ into $m_0$ disjoint datasets $D_1, \ldots, D_{m_0}$ of sizes $|D_i| = \ell_i$ as above. Then, it returns $\mathcal{A}^{(i)}(\theta_i; \bar{D}_i)$ for a dataset $\bar{D}_i$ of size $\ell_i$ constructed as follows: with probability $p_i = \mathbf{Pr}[d_1 \in \tilde{D}_i | \theta_{1:i}]$ it takes $\bar{D}_i$ to be the dataset obtained by replacing a random element from $D_i$ with $d_1$, and with probability $1 - p_i$ it takes $\bar{D}_i = D_i$. Note this construction preserves the output distribution since for any $\theta$ we have

$$\mathbf{Pr}[\mathcal{A}^{(i)}(\theta_i; \tilde{D}_i) = \theta | \theta_{1:i}] = (1 - p_i)\mathbf{Pr}[\mathcal{A}^{(i)}(\theta_i; D_i) = \theta | \theta_{1:i}, d_1 \notin \tilde{D}_i]$$
$$+ \frac{p_i}{\ell_i} \sum_{d \in D_i} \mathbf{Pr}[\mathcal{A}^{(i)}(\theta_i; D_i \cup \{d_1\} \setminus \{d\}) = \theta | \theta_{1:i}, d_1 \in \tilde{D}_i]$$
$$= \mathbf{Pr}[\mathcal{B}^{(i)}(\theta_{1:i}; D) = \theta]$$

To bound the probabilities $p_i$ we write:

$$p_i = \mathbf{Pr}[d_1 \in \tilde{D}_i | \theta_{1:i}]$$
$$= \frac{\mathbf{Pr}[\theta_{1:i}|d_1 \in \tilde{D}_i]\mathbf{Pr}[d_1 \in \tilde{D}_i]}{\mathbf{Pr}[\theta_{1:i}]}$$
$$= \frac{\ell_i}{n} \frac{\mathbf{Pr}[\theta_{1:i}|d_1 \in \tilde{D}_i]}{\sum_{k \in [m_0]} \mathbf{Pr}[\theta_{1:i}|d_1 \in \tilde{D}_k]\mathbf{Pr}[d_1 \in \tilde{D}_k]}$$
$$= \frac{\ell_i}{\sum_{k \in [m_0]} \ell_k \frac{\mathbf{Pr}[\theta_{1:i}|d_1 \in \tilde{D}_k]}{\mathbf{Pr}[\theta_{1:i}|d_1 \in \tilde{D}_i]}}$$

To proceed, we assume $\delta_0 = 0$. If that is not the case, then the same argument based on Lemma A.3 used in the proof of Theorem A.4 allows us to reduce the analysis to the case $\delta_0 = 0$ and modify the final $\varepsilon$ and $\delta$ accordingly. When the local randomizers satisfy pure DP, we have

$$\sum_{k \in [m_0]} \ell_k \frac{\mathbf{Pr}[\theta_{1:i}|d_1 \in \tilde{D}_k]}{\mathbf{Pr}[\theta_{1:i}|d_1 \in \tilde{D}_i]} \geq \ell_i + e^{-2\varepsilon_0}\sum_{k<i} \ell_k + e^{-\varepsilon_0}\sum_{k>i} \ell_k$$
$$\geq e^{-2\varepsilon_0}n$$

Thus we obtain $p_i \leq e^{2\varepsilon_0}\ell_i/n$. Now, the overlapping mixtures argument used in the proof of Theorem A.4 (see [5]) shows that $\mathcal{B}^{(i)}$ is $\varepsilon_i$-DP with $\varepsilon_i \leq \log(1 + e^{2\varepsilon_0}(e^{\varepsilon_0}-1)\ell_i/n)$. Furthermore, the heterogenous advanced composition theorem [24] implies that the composition of $\mathcal{B}^{(1)}, \ldots, \mathcal{B}^{(m_0)}$

satisfies $(\varepsilon, \delta)$-DP with

$$\varepsilon = \sum_{i \in [k]} \frac{(e^{\varepsilon_i} - 1)\varepsilon_i}{e^{\varepsilon_i} + 1} + \sqrt{2 \log \frac{1}{\delta} \sum_{i \in [k]} \varepsilon_i^2}$$

$$\leq \frac{(e^{\varepsilon_0} - 1)^2 e^{4\varepsilon_0} \|\ell\|_2^2}{2n^2} + \sqrt{\frac{2(e^{\varepsilon_0} - 1)^2 e^{4\varepsilon_0} \|\ell\|_2^2}{n^2} \log \frac{1}{\delta}}$$

$\square$

To conclude the proof of Theorem 4.1, we provide a high probability bound for $\|L\|_2$ for random $L$ representing the loads of $m$ bins when $n$ balls are thrown uniformly and independently.

**Lemma A.6.** *Let $L = (L_1, \ldots, L_m)$ denote the number of users checked in into each of $m$ update slots in the protocol from Figure 2. With probability at least $1 - \delta$, we have*

$$\|L\|_2 \leq \sqrt{n + \frac{n^2}{m}} + \sqrt{n \log(1/\delta)}.$$

*Proof.* The proof is a standard application of McDiarmid's inequality. First note that $\|L\|_2$ is a function of $n$ i.i.d. random variables indicating the bin where each ball is allocated. Since changing the assignment of one ball can only change $\|L\|_2$ by $\sqrt{2}$, we have

$$\|L\|_2 \leq \mathbb{E} \|L\|_2 + \sqrt{n \log(1/\delta)}$$

with probability at least $1 - \delta$. Finally, we use Jensen's inequality to obtain

$$\mathbb{E} [\|L\|_2] \leq \sqrt{\mathbb{E} \left[ \|L\|_2^2 \right]} = \sqrt{\sum_{i \in [m]} \mathbb{E} [L_i^2]} = \sqrt{m \mathbb{E} [\mathrm{Bin}(n, 1/m)^2]}$$

$$= \sqrt{m \left( \frac{n}{m} \left( 1 - \frac{1}{m} \right) + \frac{n^2}{m^2} \right)} \leq \sqrt{n + \frac{n^2}{m}}$$

$\square$

The privacy claim in Theorem 4.1 follows from using Lemma A.6 to condition with probability at least $1 - \delta_2$ to the case where $L$ is such that

$$\frac{\|L\|_2}{n} \leq \sqrt{\frac{1}{n} + \frac{1}{m}} + \sqrt{\frac{\log(1/\delta_2)}{n}} ,$$

and for each individual event $L = \ell$ satisfying this condition, applying the analysis from Theorem A.5 after the reduction from shuffling to averaging (see, e.g., the proof of Theorem 5.1 below).

### A.3 Proof of Theorem 5.1

---

**Algorithm 5** $\mathcal{A}_{sl}$: Local responses with shuffling

---

    **Input:** Dataset $D = d_{1:n}$, algorithms $\mathcal{A}_{ldp}^{(i)} : \mathcal{S}^{(1)} \times \cdots \times \mathcal{S}^{(i-1)} \times \mathcal{D} \to \mathcal{S}^{(i)}$ for $i \in [n]$.
1: Let $\pi$ be a uniformly random permutation of $[n]$
2: **for** $i \in [n]$ **do**
3:     $s_i \leftarrow \mathcal{A}_{ldp}^{(i)}(s_{1:i-1}; d_{\pi(i)})$
4: **return** sequence $s_{1:n}$

---

We will prove the privacy guarantee of $\mathcal{A}_{sl}$ (Algorithm 5) in a similar manner as in the proof of Theorem 7 in [19]: by reducing $\mathcal{A}_{sl}$ to $\mathcal{A}_{swap}$ that starts by swapping the first element with a u.a.r. sample in the dataset, and then applies the local randomizers (Algorithm 6). They key difference between our proof and the one in [19] is that we provide tighter, position-dependent privacy guarantees for each of the outputs of $\mathcal{A}_{swap}$, and then use an *heterogeneous* adaptive composition theorem from [24] to compute the final privacy parameters.

---

**Algorithm 6** $\mathcal{A}_{swap}$: Local responses with one swap

---

    **Input:** Dataset $D = d_{1:n}$, algorithms $\mathcal{A}_{ldp}^{(i)} : \mathcal{S}^{(1)} \times \cdots \times \mathcal{S}^{(i-1)} \times \mathcal{D} \to \mathcal{S}^{(i)}$ for $i \in [n]$.

1: Sample $I \xleftarrow{u.a.r.} [n]$
2: Let $\sigma_I(D) \leftarrow (d_I, d_2, \ldots, d_{I-1}, d_1, d_{I+1}, \ldots, d_n)$
3: **for** $i \in [c]$ **do**
4:     $s_i \leftarrow \mathcal{A}_{ldp}^{(i)}(s_{1:i-1}; \sigma_I(D)[i])$
5: **return** sequence $s_{1:n}$

---

**Theorem A.7.** *(Amplification by swapping) For a domain $\mathcal{D}$, let $\mathcal{A}_{ldp}^{(i)} : \mathcal{S}^{(1)} \times \cdots \times \mathcal{S}^{(i-1)} \times \mathcal{D} \to \mathcal{S}^{(i)}$ for $i \in [n]$ (where $\mathcal{S}^{(i)}$ is the range space of $\mathcal{A}_{ldp}^{(i)}$) be a sequence of algorithms s.t. $\mathcal{A}_{ldp}^{(i)}$ is $\varepsilon_0$-DP for all values of auxiliary inputs in $\mathcal{S}^{(1)} \times \cdots \times \mathcal{S}^{(i-1)}$. Let $\mathcal{A}_{swap} : \mathcal{D}^n \to \mathcal{S}^{(1)} \times \cdots \times \mathcal{S}^{(n)}$ be the algorithm that given a dataset $D = d_{1:n} \in \mathcal{D}^n$, swaps the first element in $D$ with an element sampled u.a.r. in $D$, and then applies the local randomizers to the resulting dataset sequentially (see Algorithm 6). $\mathcal{A}_{swap}$ satisfies $(\varepsilon, \delta)$-DP at index 1 in the central model, for $\varepsilon = \frac{e^{3\varepsilon_0}(e^{\varepsilon_0}-1)^2}{2n} + e^{3\varepsilon_0/2}(e^{\varepsilon_0}-1)\sqrt{\frac{2\log(1/\delta)}{n}}$. Furthermore, if the $\mathcal{A}^{(i)}$ are $(\varepsilon_0, \delta_0)$-DP with $\delta_0 \leq \frac{(1-e^{-\varepsilon_0})\delta_1}{4e^{\varepsilon_0}\left(2+\frac{\ln(2/\delta_1)}{\ln(1/(1-e^{-5\varepsilon_0}))}\right)}$, then $\mathcal{A}_{swap}$ is $(\varepsilon', \delta')$-DP with $\varepsilon' = \frac{e^{24\varepsilon_0}(e^{8\varepsilon_0}-1)^2}{2n} + e^{12\varepsilon_0}(e^{8\varepsilon_0} - 1)\sqrt{\frac{2\log(1/\delta)}{n}}$ and $\delta' = \delta + m(e^{\varepsilon'} + 1)\delta_1$.*

*Proof.* We start by proving the privacy guarantee of $\mathcal{A}_{swap}$ for the case where for each $i \in [c]$, the local randomizer $\mathcal{A}_{ldp}^{(i)}$ is $\varepsilon_0$-DP, i.e., for the case where $\delta_0 = 0$. Let us denote the output sequence of $\mathcal{A}_{swap}$ by $Z_1, Z_2, \ldots, Z_n$. Note that $Z_{1:n}$ can be seen as the output of a sequence of $n$ algorithms with conditionally independent randomness: $\mathcal{B}^{(i)} : \mathcal{S}^{(1)} \times \cdots \times \mathcal{S}^{(i-1)} \times \mathcal{D}^n \to \mathcal{S}^{(i)}$ for $i \in [n]$. On input $s_{1:i-1}$ and $D$, $\mathcal{B}^{(i)}$ outputs a random sample from the distribution of $Z_i | Z_{1:i-1} = s_{1:i-1}$. The outputs of $\mathcal{B}^{(1)}, \ldots, \mathcal{B}^{(i-1)}$ are given as input to $\mathcal{B}^{(i)}$. Therefore, in order to upper bound the privacy parameters of $\mathcal{A}_{swap}$, we analyze the privacy parameters of $\mathcal{B}^{(1)}, \ldots, \mathcal{B}^{(n)}$ and apply the heterogeneous advanced composition for DP [24].

Next, observe that conditioned on the value of $I$, $Z_i$ is the output of $\mathcal{A}_{ldp}^{(i)}(s_{1:i-1}; d)$ with its internal randomness independent of $Z_{1:i-1}$. In particular, for $i \geq 2$, one can implement $\mathcal{B}^{(i)}$ as follows. First, sample an index $T$ from the distribution of $I | Z_{1:i-1} = s_{1:i-1}$. Output $\mathcal{A}_{ldp}^{(i)}(s_{1:i-1}; d_1)$ if $T = i$, otherwise output $\mathcal{A}_{ldp}^{(i)}(s_{1:i-1}; d_i)$. For $\mathcal{B}^{(1)}$, we first sample $T$ u.a.r. from $[n]$, and then output $\mathcal{A}_{ldp}^{(1)}(d_T)$.

We now prove that for each $i \in [c]$, $\mathcal{B}^{(i)}$ is $\left(\log\left(1 + \frac{e^{2\varepsilon_0}(e^{\varepsilon_0}-1)}{e^{2\varepsilon_0}+(i-1)+(n-i)e^{\varepsilon_0}}\right), 0\right)$-DP at index 1. Let $D = d_{1:n}$ and $D' = (d_1', d_{2:n})$ be 2 datasets differing in the first element. Let $s_{1:i-1}$ denote the input to $\mathcal{B}^{(i)}$. Let $\mu$ be the probability distribution of $\mathcal{B}^{(i)}(s_{1:i-1}; D)$, and let $\mu_0$ (resp. $\mu_1$) be the distribution of $\mathcal{B}^{(i)}(s_{1:i-1}; D)$ conditioned on $T \neq i$ (resp. $T = i$). Let $q_i$ be the probability that $T = i$ (sampled from $I | Z_{1:i-1} = s_{1:i-1}$). By definition, $\mu = (1 - q_i)\mu_0 + q_i\mu_1$. Also, denote by $\mu', \mu_0', \mu_1'$, and $q_i'$ the corresponding quantities when $\mathcal{B}^{(i)}$ is run on $D'$. Thus, we get $\mu' = (1 - q_i')\mu_0' + q_i'\mu_1'$.

For $i \in [n]$, we observe that $\mu_0 = \mu_0'$, since in both cases the output is generated by $\mathcal{A}_{ldp}^{(i)}(d_T)$ conditioned on $T \neq 1$ for $i = 1$, and $\mathcal{A}_{ldp}^{(i)}(s_{1:i-1}; d_i)$ for $i \geq 2$. W.l.o.g. assume that $q_i \geq q_i'$. Thus, we can shift $q_i - q_i'$ mass from the first component of the mixture in $\mu'$ to the second component to obtain

$$\mu' = (1 - q_i)\mu_0 + q_i\left(\frac{q_i'}{q_i}\mu_1' + \left(1 - \frac{q_i'}{q_i}\right)\mu_0\right) = (1 - q_i)\mu_0 + q_i\mu_1''$$

This shows that $\mu$ and $\mu'$ are overlapping mixtures [5]. Now, $\varepsilon_0$-LDP of $\mathcal{A}_{ldp}^{(i)}$ implies $\mu_0 \cong_{(\varepsilon_0,0)} \mu_1$ and $\mu_0 \cong_{(\varepsilon_0,0)} \mu_1'$. Moreover, $\varepsilon_0$-LDP of $\mathcal{A}_{ldp}^{(i)}$ also implies $\mu_1 \cong_{(\varepsilon_0,0)} \mu_1'$, so by the joint convexity of the relation $\cong_{(\varepsilon_0,0)}$ we also have $\mu_1 \cong_{(\varepsilon_0,0)} \mu_1''$. Thus, we can apply Advanced Joint Convexity of overlapping mixtures (Theorem 2 in [5]) to get that

$$\mu \cong_{(\log(1+q_i(e^{\varepsilon_0}-1)),0)} \mu' \tag{13}$$

We now claim that $q_i \leq \frac{e^{2\varepsilon_0}}{e^{2\varepsilon_0}+(i-1)+(n-i)e^{\varepsilon_0}}$. Observe that for each $D^* \in \{D, D'\}$, conditioning on $T = i$ reduces $\mathcal{A}_{swap}$ to running $\mathcal{A}_{ldp}^{(k)}, k \in [n]$ on $\sigma_i(D^*)$. Note that $\sigma_i(D^*)[1:i-1]$ differs from $\sigma_j(D^*)[1:i-1]$ in at most 2 positions for $j < i$, and at most 1 position for $j > i$. By $\varepsilon_0$-LDP of $\mathcal{A}_{ldp}^{(k)}, k \in [n]$, we get that

$$\frac{\mathbf{Pr}[Z_{1:i-1}=s_{1:i-1}|T=i]}{\mathbf{Pr}[Z_{1:i-1}=s_{1:i-1}|T=j]} \leq e^{2\varepsilon_0} \text{ for } j<i \quad \text{and} \quad \frac{\mathbf{Pr}[Z_{1:i-1}=s_{1:i-1}|T=i]}{\mathbf{Pr}[Z_{1:i-1}=s_{1:i-1}|T=j]} \leq e^{\varepsilon_0} \text{ for } j>i \tag{14}$$

Now, on the lines of the proof of Lemma A.1, we have:

$$\frac{\mathbf{Pr}[Z_{1:i-1}=s_{1:i-1}|T=i]}{\mathbf{Pr}[Z_{1:i-1}=s_{1:i-1}]}$$

$$= \frac{\mathbf{Pr}[Z_{1:i-1}=s_{1:i-1}|t=i]}{\sum_{j=1}^{n} \mathbf{Pr}[Z_{1:i-1}=s_{1:i-1}|T=j]\mathbf{Pr}[T=j]}$$

$$= \frac{1}{\sum_{j=1}^{n} \frac{\mathbf{Pr}[Z_{1:i-1}=s_{1:i-1}|t=j]}{\mathbf{Pr}[Z_{1:i-1}=s_{1:i-1}|t=i]} \mathbf{Pr}[T=j]}$$

$$= \frac{n}{1+(i-1)\sum_{j<i}\frac{\mathbf{Pr}[Z_{1:i-1}=s_{1:i-1}|T=j]}{\mathbf{Pr}[Z_{1:i-1}=s_{1:i-1}|T=i]}+(n-i)\sum_{k>i}\frac{\mathbf{Pr}[Z_{1:i-1}=s_{1:i-1}|T=k]}{\mathbf{Pr}[Z_{1:i-1}=s_{1:i-1}|T=i]}}$$

$$\leq \frac{n}{1+(i-1)e^{-2\varepsilon_0}+(n-i)e^{-\varepsilon_0}} = \frac{ne^{2\varepsilon_0}}{e^{2\varepsilon_0}+(i-1)+(n-i)e^{\varepsilon_0}}$$

where the third equality follows as for every $j \in [n], \mathbf{Pr}[T=j]=\frac{1}{n}$, and the first inequality follows from inequality 14.

This immediately implies our claim, since

$$q_i = \mathbf{Pr}[T=i|Z_{1:i-1}=s_{1:i-1}] = \frac{\mathbf{Pr}[Z_{1:i-1}=s_{1:i-1}|T=i] \cdot \mathbf{Pr}[T=i]}{\mathbf{Pr}[Z_{1:i-1}=s_{1:i-1}]}$$

$$\leq \frac{e^{2\varepsilon_0}}{e^{2\varepsilon_0}+(i-1)+(n-i)e^{\varepsilon_0}}$$

where the inequality follows from (11), and as $\mathbf{Pr}[T = i] = \frac{1}{n}$. Substituting the value of $q_i$ in (13), we get that for each $i \in [n]$, algorithm $\mathcal{B}^{(i)}$ is $(\varepsilon_i, 0)$-DP at index 1, where $\varepsilon_i = \log\left(1+\frac{e^{2\varepsilon_0}(e^{\varepsilon_0}-1)}{e^{2\varepsilon_0}+(i-1)+(n-i)e^{\varepsilon_0}}\right)$. This results in $\varepsilon_i \leq \log\left(1+\frac{e^{\varepsilon_0}(e^{\varepsilon_0}-1)}{n-(i-1)\left(1-\frac{1}{e^{\varepsilon_0}}\right)}\right)$, and using Lemma A.2 for the sequence of mechanisms $\mathcal{B}^{(1)}, \ldots, \mathcal{B}^{(n)}$ by setting $a = e^{\varepsilon_0}(e^{\varepsilon_0}-1)$, $b = 1-\frac{1}{e^{\varepsilon_0}}$, and $k = n$, we get that algorithm $\mathcal{A}_{swap}$ satisfies $(\varepsilon, \delta)$-DP at index 1, for $\varepsilon = \frac{e^{3\varepsilon_0}(e^{\varepsilon_0}-1)^2}{2n} + e^{3\varepsilon_0/2}(e^{\varepsilon_0}-1)\sqrt{\frac{2\log(1/\delta)}{n}}$.

The case $\delta_0 > 0$ uses the same argument based on Lemma A.3 used in the proof of Theorem A.4. This arguments allows us to reduce the analysis to the case $\delta_0 = 0$ and modify the final $\varepsilon$ and $\delta$ accordingly.

$\square$

Now, we are ready to prove Theorem 5.1.

*Proof of Theorem 5.1.* This proof proceeds in a similar manner as the proof of Theorem 7 in [19]. Let $D$ and $D'$ be 2 datasets of length $n$ that differ at some index $i^* \in [n]$. Algorithm $\mathcal{A}_{sl}$ can be alternatively seen as follows. Pick a random one-to-one mapping $\pi^*$ from $\{2, \ldots, n\} \to [n] \setminus \{i^*\}$ and let $\pi^*(D) = (d_{i^*}, d_{\pi^*(2)}, \ldots, d_{\pi^*(n)})$. Next, apply $\mathcal{A}_{swap}$ to $\pi^*(D)$. It is easy to see that for a u.a.r. chosen $\pi^*$ and u.a.r. $I \in [n]$, the distribution of $\sigma_I(\pi^*(D))$ is a uniformly random permutation of elements in $D$.

For a fixed $\pi^*$, we know that $\pi^*(D)$ and $\pi^*(D')$ differ only in the element with index 1. Therefore, in the case $\delta_0 = 0$, from Theorem A.7, we get $\mathcal{A}_{swap}(\pi^*(D)) \cong_{\varepsilon, \delta} \mathcal{A}_{swap}(\pi^*(D'))$ at index 1, for $\varepsilon = \frac{e^{3\varepsilon_0}(e^{\varepsilon_0} - 1)^2}{2n} + e^{3\varepsilon_0/2}(e^{\varepsilon_0} - 1)\sqrt{\frac{2\log(1/\delta)}{n}}$, which implies $\mathcal{A}_{sl}(D) \cong_{\varepsilon, \delta} \mathcal{A}_{sl}(D')$.

The case $\delta_0 > 0$ follows similarly from the corresponding setting of Theorem A.7. $\qquad\square$