[Reviews · NeurIPS 2020]

Review 1

Summary and Contributions: Differentially Private - Stochastic Gradient Descent (DP-SGD) is (arguably) the only known differentially private algorithm for training deep learning models. However, DP-SGD algorithm is hard to implement in practice (even in centralized setting), but becomes almost prohibitive in distributed settings such as Federated learning. The paper proposes a natural protocol, (Random check-in), for making DP-SGD distributed.

Strengths: The trouble with DP-SGD is that it assumes that every batch (minibatch, lot etc) is a uniform sample of all the users. This seeming innocuous assumption is notoriously hard to implement even in centralized setting. On the other hand, it is crucial in the analysis; Without this assumption, guarantees of DP-SGD are useless. It is not hard to see that this assumption is impractical to implement in FL setting as all the users may not be even available at all times. Strengths: 1. Problem studied is important: DP-SGD is one of the most important algorithms in private ML, and understanding how to implement in distributed setting is an important problem. 2. Proposed solution is natural. Moreover, it seems reasonable enough that it can be implemented in practice. 3. Bounds obtained almost match the guarantees of DP-SGD in the central model.

Weaknesses: Paper does not introduce (m)any new mathematical techniques.

Correctness: Yes.

Clarity: The paper is easy to follow.

Relation to Prior Work: yes.

Reproducibility: Yes

Additional Feedback: I do not know. I am not sure FL model can be reproduced. But random-check in assumption can be simulated perhaps.


Review 2

Summary and Contributions: The primary contribution of this paper is to use the concept of randomized check-in by the participants to amplify the differential privacy guarantees of a federated learning scheme with a trusted aggregator. Similar to privacy amplification by sub-sampling, each model update is based on the averaged "noisy" gradient from a subset of participants. However, the key novelty is that the participants have the power to independently decide whether to contribute to a particular update instead of the server picking a random subset. Updates after response: Overall, I agree that the paper has addressed an important problem and made some novel theoretical contributions. The primary motivation of this work is that current amplification techniques are not practical from a privacy standpoint. My main concern is that the proposed approach goes in the opposite direction and makes it impractical from the utility standpoint. Which is why I'm hesitant to whole-heartedly support the paper and I retain my original review rating.

Strengths: 1) The main strength of the paper is that it provides a more practical approach for privacy amplification via sub-sampling. In the traditional server-mediated sub-sampling, the participants chosen by the server may not be available to provide a gradient update during the requested time window. In contrast, in the proposed approach, the client randomly commits to provide an update at a specific time slot if requested by the server. 2) The paper also proves that the above approach achieves roughly the same order of privacy amplification as that of sub-sampling/shuffling. It also provides a clear comparison of the privacy amplification achieved by related schemes.

Weaknesses: There are five main problems with the proposed scheme: 1) It is not clear what will happen to the privacy and utility guarantees if the participant J_i selected by the server for a particular time slot fails to provide a gradient update within the specified time window? 2) The privacy guarantees are based on the strict assumptions of a trusted server and privileged communications between the participant and the server, which effectively hides the information of the subset of clients participating in each update. However, just by examining the communication patterns of the server, an adversary may be able infer the subset of participants involved in a specific update. This is especially dangerous in Algorithm 1, where only a single participant's gradient is used to update the model. Thus, the gradient update of that participant may get leaked. 3) It is not clear why the so-called utility guarantee has been claimed to be optimal. Even the guarantee in Theorem 3.5 is only for convex ERM. Does that imply that the proposed algorithm is unlikely to work in typical deep neural network training scenarios? 4) There is no analysis (theoretical or empirical) of the convergence properties of the proposed scheme. 5) The threat caused by malicious participants could be higher in the proposed approach compared to other approaches, because a malicious participant now has the power to decide when to participate in a model update. Updates after response: The authors have tried to address some of the above issues in their response. a) They do agree that drop-outs will have an impact on the utility. b) I agree that assumptions of a trusted server and privileged communications are required for most existing amplification guarantees (e.g., by sampling or shuffling). However, none of the existing methods rely on aggregated updates from a small subset of participants (worst case is a single participant). In the worst case, only the LDP guarantee remains and all the advantages of aggregation are lost. c) I agree that convergence analysis for non-convex settings is a hard problem. Which is why some kind of empirical results showing that the proposed approach works in practice is essential. There is no empirical evidence/experiment to show that the proposed approach works in practice (e.g., for federated deep learning) and what are the practical convergence issues. d) I agree that malicious clients are unlikely to break the privacy guarantees, but they can easily provide corrupted updates that can ruin the learning process. In other words, malicious users can easily degrade the utility because they have more power to mount a targeted attack on the proposed approach.

Correctness: The differential privacy claims appear to be intuitively correct, even though I'm not sure if all the proofs in the supplementary material are accurate. There is no empirical evaluation of the proposed approach.

Clarity: Yes, the paper is reasonable well-written. One area of improvement is that the current focus is mainly on Algorithm 1 and its analysis. Maybe the benefits of the other variations in Sections 4 and 5 could be better explained and compared with the approach in Section 3.1. Furthermore, the paper appears to end abruptly.

Relation to Prior Work: Yes, the relationship to previous work has been clearly explained and the benefits of the proposed approach have been highlighted.

Reproducibility: Yes

Additional Feedback:


Review 3

Summary and Contributions: This paper studies privacy amplification for differentially private stochastic gradient descent (DP-SGD) in the setting of federated learning. Specifically, it proposes a new distributed protocol, termed "random check-in", which allows each client to decide on their participation locally and independently. In comparison, privacy amplification in prior work assumes that the data can be uniformly sampled (each client provides an update when requested). The paper formally analyses the privacy guarantees of three random check-in protocols and shows that for one of these settings, the utility guarantees also match the optimal privacy/accuracy trade-offs for convex DP-SGD in the central DP model. Lastly, as a byproduct of the analysis, the authors give an improved analysis of the known privacy amplification by shuffling technique, by improving the dependence of the final privacy parameter on the local randomizers' parameter, and by extending it to the case of approximate-DP local randomizers.

Strengths: -The paper explores DP-SGD in more practical federated learning settings where communication between the trusted server and the clients is initiated by the clients. The authors provide a thorough analysis of three interesting instantiations of their proposed model of random check-ins, including a setting where the clients are available during small windows of time with respect to the duration of the protocol. -The authors prove that one of these instantiations matches the optimal privacy/accuracy trade-offs known for the case of convex SGD under central DP, for the right setting of parameters of their protocol, providing useful insights about this new model of communication. -The improved analysis and the extension of privacy amplification via shuffling to approximate DP are interesting in their own right.

Weaknesses: -One could argue that the instantiations of the proposed distributed protocol are still imposing a lot of structure to the availability of the clients, e.g. each client picks uniformly at random in which time step to participate, or they are available for fixed size windows which are largely overlapping, and thus do not capture practical settings. However, I think that a) the latter model of clients "waking up in order" still captures interesting settings and b) this is already a big step in the analysis of these asynchronous settings, which required interesting technical ideas.

Correctness: The claims are correct and fully proven.

Clarity: I think that the presentation of the paper is excellent, especially the main body.

Relation to Prior Work: Yes.

Reproducibility: Yes

Additional Feedback: I really liked the presentation of the paper. -My understanding is that the probability p_j of participating is known to the algorithm (to set up the step eta) but the slot R_j is not, correct? I think it would be useful to mention that somewhere in Section 3. -I would have liked a discussion on --the utility of the sliding window algorithm (The discussion on how to set the parameters m, p_0, to retrieve and compare with the standard utility bounds for n updates in the fixed window setting was useful. Perhaps, discuss the case of constant m?) and --the importance of this imposed structure of overlapping slots (you mention it is there for simplicity, but what is the difficulty when this is not the case?). Minor typos: -Footnote 6: delta<1/100 -Proof of Theorem 3.5: 1/(1-p_b)<=1/(1-p') ================================ Thank you for your response. I think it would be nice for the reader to add some of these examples you mentioned for the settings you have chosen to analyze, and have a note about R_j. I support acceptance of this paper, mainly based on the fact that it makes the first steps towards exploring more realistic models of private distributed learning.

[Author Response · NeurIPS 2020]

We sincerely thank all the reviewers for their thorough feedback, and detailed comments. We will incorporate all feedback related to presentation (typos, stating benefits of variations, ending abruptly, etc.) in the draft. We'd like to start by addressing the feedback of reviewer 2:

- *If a selected participant fails to provide gradient update*: For clarity of the framework, we have not modelled various kinds of 'drop-outs' that could take place in such a system, e.g., dropping out after a random check-in, disconnecting after receiving the model from the server, etc. Our framework is designed such that the privacy guarantees will not degrade due to such drop-outs. One way to envision this is a time-based system, where the server stops waiting for some client after a predetermined amount of time, and proceeds with applying an all-zero model update with noise. Thus, the utility of the system will depend on such factors, but not privacy.

- *Assumptions of a trusted server and privileged communications*: The primary focus of our work is to ensure a model published to the world doesn't regurgitate private data, for e.g., a language model accurately completing the sentence "John Doe's credit card number is ...". Requiring less trust from the server is also an interesting problem for a distributed setup, and addressing this may be best accomplished by orthogonal techniques like Secure Multi-Party Computation. Moreover, in our framework each model update sent from a client device does obtain a local differential privacy (LDP) guarantee, which is what gets amplified for a central DP guarantee. The amplified guarantees do assume both the stated assumptions, however we would like to state that these assumptions hold for all existing amplification guarantees for a general distributed learning setting (namely, privacy amplification by sampling and shuffling).

- *No analysis of convergence; why optimal ... unlikely to work in NN training?*: We do state that the utility guarantees of our main protocol are *optimal for convex ERMs*, and we also provide a bound on the number of "dummy" updates in general. The method of DP-SGD is commonly used in practice for training deep NNs (without any formal utility guarantees), and our technique is modeled on that. Moreover, for non-convex settings, very little is known in general about optimality/convergence in the DP literature.

- *Threat by malicious clients could be higher due to local randomness*: The assumption of no malicious clients in our setup is required only by the "thrifty" updates version of the algorithm (Section 4.1), and the privacy guarantees degrade smoothly with the proportion of malicious users (similar to the analyses of privacy amplification via shuffling). For the other two versions of our algorithm (the main version in Section 3.1, and the sliding window version in Section 4.2), the amplified privacy of a client device is crucially dependent on that client and the server following the protocol, and thus, it won't degrade by the actions of malicious clients. In other words, the central DP guarantee provided is for all clients that follow the protocol along with the server.

- *Other variations could be compared with the approach in Section 3.1*: Since the submission deadline, we have worked out a comparison of the utility of the algorithm in Section 4.1 to the main algorithm in Section 3.1, in that for convex non-smooth losses and $m \ll n$, the main algorithm provides a better utility whereas the utility bounds are incomparable for smooth losses.

Addressing the feedback of reviewers 1 and 2 regarding empirical evaluation of random check-ins: There are various design choices to be made for modeling real-world device availability (e.g., diurnal variations, overlap in availability for small/large/global populations, device capabilities across the population, etc.) to enable running appropriate simulations, and thus we leave it for future work.

Addressing the feedback of reviewer 3:

- *Practicality of local u.a.r. slot selection, and availability for largely overlapping fixed-size windows*: The design choices for our framework are motivated by real-world applications, such as a client device can actually locally determine and participate in a u.a.r. slot in its own availability window (since it is available to participate during each instant of its availability window). Similarly, federated learning involving a population of a country, for instance, can be expected to have largely overlapping windows due to diurnal variations in client device availability. A sliding-window type behavior can be expected for learning from a global population (due to shifting time-zones for various sub-populations).

- *Slot $R_j$ known to the algorithm?*: The amplified DP guarantees are central DP guarantees, which assume a trusted server, and thus $R_j$ is known to the algorithm (though it cannot be released publicly for the amplification, similar to 'secrecy of the sample' in amplification due to sampling, or the random ordering being secret in amplification due to shuffling).

- *Difficulty for non-overlapping slots*: The privacy guarantees of our protocols do not depend on any overlapping structure of the slots, but the utility of a protocol will have a dependence. Thus, we analyze some variants that were motivated by practical applications, such as limited overlap of clients that is motivated by diurnal variations in training on local/national populations, and sliding window availability additionally motivated by shifting time-zones for training on a global population.

[Meta-Review · NeurIPS 2020]

This paper provides a new privacy amplification technique for differential privacy. The proposed solution is a novel contribution to the literature. During the reviewer discussion, there were concerns that the method is not fully practical and yet the main motivation of the paper is that the previous techniques may not work in practice when the data are distributed across different nodes. The authors should provide an honest discussion of these practical issues raised in the reviews.